



# Total Air Content measurements from the RECAP ice core

Sindhu Vudayagiri[1], Bo Vinther[1], Johannes Freitag[2], Peter L. Langen[3], Thomas Blunier[1]*,

1 Physics of Ice, Climate and Earth, Niels Bohr Institute, University of Copenhagen, Tagensvej 16, 2200 Copenhagen, Denmark

5 2 Alfred Wegner Institute, Snow and Firn Section Glaciology, 27570 Bremerhaven, Germany

Department of Environmental Science, iClimate, Aarhus University, Frederiksborgvej 399, 4000 Roskilde, Denmark

*Correspondence to: Thomas Blunier (blunier@nbi.ku.dk), Tel: +45 35 32 05 84

**Abstract**. Total air content (TAC) of the REnland ice CAP project (RECAP) core, drilled in summer 2015, is measured as a part of investigating the elevation history of the Greenland Ice Sheet (GIS). TAC is a proxy for the elevation at which

the ice was originally formed as the TAC in ice cores is predominantly influenced by surface air pressure and conditions like temperature and local summer insolation. The RECAP TAC data shows incoherently low values in the Holocene climatic optimum (6 to 9 kyr b2k) and in the Eemian (119 to 121 kyr b2k) which renders the TAC data unfit for paleo elevation interpolation. In contrast, the glacial section (11.7 kyr to 119 kyr b2K) has consistent TAC values thus in principle facilitating the past elevation calculations. However, we observe TAC variations related to Dansgaard-Oeschger events (D-O) that cannot originate from elevation changes but must be linked to changes in the firn structure. We analyse

the pattern of the structural changes in the RECAP and NGRIP cores. For the melt affected sections (Holocene and Eemian) we use melt affected TAC to reconstruct summer temperatures.

*Keywords*: Greenland Ice sheet, Renland, Total air content, elevation change, Insolation, melt layers.

## 1 Introduction

We present the first total air content (TAC) record from the Renland ice cap from the RECAP core, drilled in 2015. The TAC data from RECAP is of particular interest as it captures the Eastern Greenland atmospheric conditions along with the elevation history (Raynaud et al., 1982) of the Renland ice cap. The current study was designed to answer the pertinent question if the Renland ice cap always had the same elevation. Vinther et al (2009) estimated the elevation changes

induced in the GIS after the onset of the Holocene from the differences in the $\delta^{18}$O signals of GRIP, NGRIP, Camp Century and DYE-3. They found that though Greenland experienced fairly uniform climatic conditions during the Holocene, the response of the GIS has been erratic at different locations. The study uses $\delta^{18}$O of $H_2O$ data from the Renland ice cap as an anchor point arguing that the ice cap has not experienced significant ice flow or elevation changes due to its isolation from the GIS owing to the surrounding topography. The TAC signal from the RECAP core could be

used to infer the elevation changes thereby supporting or refuting this assumption. However, we learned that the RECAP TAC data is affected by melt during the Holocene and the melt fractions in the samples are construed by assuming a linear relationship between the TAC of a sample and the melt fraction of the sample. The effect of local summer insolation on the RECAP TAC signal is analysed. The RECAP TAC signal from the glacial section, unaffected by melt, is, however, affected by rapid climate change events that hinder reconstruction of the past elevation of the Renland ice cap except for

the climatically stable phase of the Last Glacial Maximum.

### 1.1 Renland ice cap

The Renland ice cap is situated in Eastern Greenland on a high elevation plateau on the Renland peninsula in the Scoresbysund fjord (Fig. 1) with a present elevation of 2340 m a.s.l. at the summit. The RECAP core is ~584 m long and was drilled to bedrock in 2015 at an elevation 2315 m a.s.l. (71° 18' 18" N; 26° 43' 24" W) near summit. The present day





annual mean temperature is -18°C measured in the firn, and the present accumulation rate is approximately 0.5 m of ice equivalent precipitation per year. In the interior of the Greenland ice sheet the average size of the air bubbles is monotonically decreasing with depth till they disappear forming clathrates (Shoji and Langway, 1982). The enclosed air in the RECAP core exists fully in the form of air bubbles. At the given temperature, clathrate formation would start below the bedrock depth of ~584 m below surface (Uchida and Hondoh, 2000). The bubble diameter, however, is increasing

again from about 530m below the surface (supplemental Fig. S3). This may be due to fast thinning of the annual layers in the Renland ice cap causing the small bubbles to coalesce to form bigger bubbles.

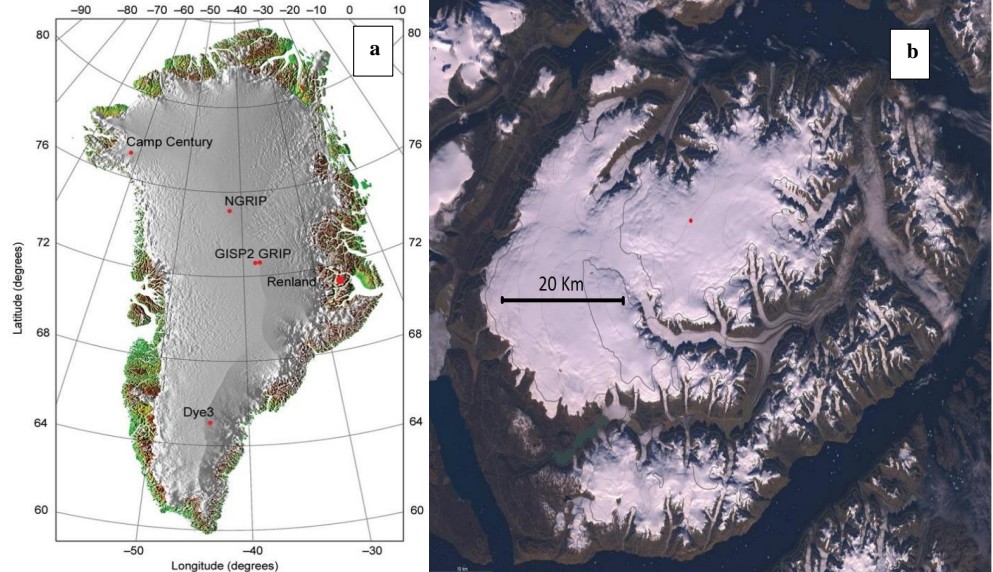

**Figure 1: (a) Map of Greenland, showing the location of the Renland ice cap and other cores (Danish Cadastre) (b) Satellite image of the Renland peninsula, which is almost entirely covered by the Renland ice cap (Landsat).**


### 1.2 Total air content in ice

The density of dry sintered snow at the surface of an ice sheet is typically 0.3–0.35 g cm$^{-3}$. This open porous firn is then densified by compaction and dry sintering to a density of 0.81–0.84 g cm$^{-3}$ where the open pores are isolated. The amount of gas trapped at this time, the total air content (TAC) depends on the pore volume, the temperature, and the pressure (e.g.

Martinerie et al., 1992). It is usually expressed as cm$^3$ of gas per kg of ice at standard temperature and pressure (STP) (equation 1) where $V_c$, $P_c$, and $T_c$ are pore volume per kg of ice, and pressure, temperature at close respectively. P and T are standard temperature and standard pressure (1013 mbar and 273.15 K).

$$TAC = V_c \frac{P_c}{T_c} \frac{T}{P} \tag{1}$$

With known $V_c$ and $T_c$ elevation changes can be obtained applying the barometric formula (equation 2).

$$P_c = P_a \left[ \frac{T_a}{T_c} \right]^{\frac{gM_{air}}{R(\frac{dT}{dz})}} \text{ with } T_c = T_a + \frac{dT}{dz}(h_c - h_a) \tag{2}$$

Where $P_c$ is the pressure at altitude $h_c$, $P_a$, $T_a$, and $h_a$ are pressure, temperature, and elevation at sea level, respectively. $dT/dz$ is the lapse rate at the location, $M_{air}$ the molecular mass of air, g the gravitational constant, and R the gas constant. A 1% change in TAC at the elevation of RECAP corresponds to a pressure change of about 7 mbar and 80m change in elevation.





A limitation of tracking elevation changes with TAC is variations in $V_c$. The variation of pore volume ($V_c$) at air isolation depth with temperature ($T_c$) has been studied in detail based on data from Antarctic and Greenland from sites which do not show summer melting (Martinerie et al., 1994). With a correlation coefficient of 0.90 the following linear relation has been found:

$$V_c \, (\text{cm}^3 \, \text{g}^{-1}) = 6.95 \times 10^{-4} \, T_c(K) - 0.043 \qquad (3)$$

We will apply this parametrization to calculate the pore volume for the RECAP core as it is based on a wide dataset from sites with a temperature range ~-15 to -60°C. Addition of two more sites has minimally change this parametrization (Delmotte et al., 1999).

At any site short term variability of $V_c$ is observed. It is explained by the variability of the density originating from
summer to winter precipitation and successive metamorphosis throughout the firn column to the air insolation depth (Hörhold et al., 2011). High density wind crusts potentially add to the variability as a minor dependence of $V_c$ to wind speed has been found (Martinerie et al., 1994). The short-term variability is on the order of 20%.

Solar insolation is mostly absorbed in the upper 2 cm of the polar ice sheet and influences ice properties leading to $O_2/N_2$ variations, induces surface warming, evaporation and formation of surface hoars (Brandt and Warren, 1993; Bender,
2002). The consistent variation of $O_2/N_2$ ratio in the air trapped in ice cores with local summertime insolation was first identified in the Vostok ice core record (Bender, 2002). The $O_2/N_2$ ratio of GISP2 is in antiphase with local summer insolation, which is consistent with findings from east Antarctic cores like Vostok and Dome Fuji (Suwa and Bender, 2008; Kawamura et al., 2007). The influence of local summer insolation on TAC signal and the anti-correlation between the TAC and insolation signal was first identified from the air content records of Antarctic plateau (Raynaud et al., 2007)

The absorbed insolation induces considerable surface warming, reflected for example in diurnal temperature variations of up to 20ºC in the surface of the ice sheet summit, Greenland (Alley et al., 1990). Summer time insolation influences the metamorphism of snow near the surface of polar ice as it causes evaporation and grain growth (Bender, 2002). It is explained that summer insolation causes rapid grain growth in the snow surface by creating an apparent summer temperature gradient. Thus, the increase in grain size below the surface affects the densification process. An increase in
insolation thereby causes the grain size to increase, porosity at close-off to decrease and density at close off to increase. The proposed mechanism explains the anti-correlation between the integrated summer insolation and the TAC, as insolation increases, porosity at close off and pore volume decreases, causing an overall decrease of the TAC. After correcting for the effect of changing local solar insolation, TAC can be interpreted to give paleo surface elevations, with high TAC corresponding to lower elevations (Raynaud et al., 1997; Raynaud et al., 2007; NEEM Community Members,
2013). Similar to the Antarctic ice cores (Raynaud et al., 2007), NGRIP TAC signal also exhibits anti-correlation with local summer insolation (>320 W/m[2]) (Eicher et al., 2016). The TAC and $O_2/N_2$ signal from Vostok, a low accumulation site further confirms that insolation has a profound influence on these signals to the extent that these signals can be used as reliable proxies for local insolation and hence can be used for orbital dating of ice cores despite the remaining gaps in our understanding of the physical mechanisms (Lipenkov et al., 2011). Interestingly, the $O_2/N_2$-insolation relation is
comparable at low accumulation sites like Vostok and high accumulation site like GISP2 with present day accumulations rates of 2.2 cm and 23 cm w. eq., respectively. Suwa and Bender (2008) speculate that either the sensitivity of the ice properties to local summer insolation is similar for a wide range of temperatures and accumulation rates or, these properties are more sensitive to insolation under higher temperature, but the effect is then attenuated by the higher accumulation rate.

Furthermore, TAC has also been found to be influenced by rapid climatic transitions in connection with Dansgaard-Oeschger events during the last glacial (Eicher et al., 2016), hence TAC measurements from climatically stable periods are preferable for past elevation estimation.





## 2 Measurements

Measurements of the RECAP ice core were made at PICE and PSU. While the system at PICE is dedicated to total air
content measurements following the barometric method and giving absolute calibrated volumes (Lipenkov et al., 1995),
the measurements at PSU are a by-product of measurements for $\delta^{15}$N and $CH_4$ contents.

### 2.1. TAC measurements at PICE

The set up consists of 3 extraction chambers connected to a differential pressure gauge via an extraction line (¼" stainless
steel tube) that is separated into various sections by stainless steel bellows sealed valves (Swagelok, SS-4H) (Aagaard,
2015). The samples are placed in extraction chambers that can be vacuum sealed and attached to the extraction line via
Swagelok VCR fittings. The extraction line is under vacuum which is maintained with the help of a vacuum pump
(Pfeiffer DUO 3 M, DN 16 KF). The differential pressure gauge used is P-BADP/P-BADR, Smart/HART pressure
transmitter that can measure a differential pressure of 0.1 KPa to 10 MPa with an accuracy of ± 0.075%. The extraction
chambers are made of aluminium with outer dimensions of 42 x 60 x 60 mm and inner dimensions of 32 x 32 x 32 mm.
The chamber is designed with rounded edges and the bottom floor inside the chamber has 3 little bumps of 1mm in height
to prevent any air from getting trapped below the ice sample. The chambers can be cooled or heated from the bottom
upwards by placing them on a base fitted with Peltier plates. The extraction line has a water vapor trap followed by a
Haysep trap (Haysep D 20/40 mesh). An extra volume is also provided in the measuring area for increasing the measuring
volume if needed.
Cubical samples (~ 22 x 25 x 25 mm), weighing ~ 10 to 15 g each are cut at specific depths of the ice core (supplemental
picture S1). The cut samples are then photographed, weighed and the dimensions measured accurately. Each ice sample
is placed inside a pre-cooled chamber and sealed airtight by fastening the 8 screws on the lid along with an O ring (NBR
70) between the lid and the chamber. The chambers are then connected to the set up via VCR connections and kept cold
by the Peltier plates upon which they are placed. The chambers with ice samples are evacuated for 30 minutes. Then the
chambers are sealed off and the ice samples are melted and re-frozen with the help of the Peltier plate. The gas released
from the sample is transferred through a water trap cooled by dry ice onto a Haysep trap held at $LN_2$ temperature. A small
percentage of gas remains dissolved in the frozen meltwater. Therefore, another melt-refreezing cycle is used to collect a
maximum of the gas from the sample. By heating the Haysep trap, the captured gas is released into the calibrated
measuring volume and its pressure is measured by the differential pressure gauge together with the room temperature.
Calibration of the differential pressure gauge and the measuring volume are briefed in the supplementary information.

### 2.2 TAC measurements at PSU

Two sets of TAC measurements are obtained at PSU. The samples used for $CH_4$ measurements are cylinders of diameter
4.1cm, height of 5.5±0.3cm, weighing 65 ± 3g each and the samples used in $\delta^{15}$N measurements are rectangular cubes of
ice (2 x 1.2 x 5cm) weighing ~13g each. In both these measurements, an automatic air extraction device (referred to as
"The Spider"), which employs the vacuum volumetric principle is used. The volume of the extracted air is measured after
which the air samples are used for $CH_4$ and $\delta^{15}$N measurements (Fegyveresi, 2015).
The spider apparatus consists of 14 steel vessels used to hold ice samples, each with a total sampling volume of
~96 ± 2 cm³ (Fegyveresi, 2015). During measurements, the system performs a single melt-refreeze cycle to free the
trapped air from within the ice (Fegyveresi, 2015). Ice samples are placed in the respective vessels and isolated from the
ambient atmosphere using copper gaskets. The entire system is then evacuated to 0.3 mbar to remove air in each vessel's
head-space, and various leak-checks are performed to ensure the seals are intact with no contamination from ambient air.
The ice samples are then melted allowing the air trapped in the ice samples to escape into the headspace of the enclosing
vessels. The melt is then refrozen, leaving the liberated air separated above each of the refrozen samples. Once the





temperature of the ice reaches -69°C, the air in each vessel is expanded into a vacuum manifold containing a 10 cm$^3$

sample loop, which is then connected to a gas chromatograph (Fegyveresi, 2015). The pressure in the vacuum manifold

with the ice core air sample is noted (generally between 60 and 80 torr) before the loop is switched for CH$_4$ concentration

or δ$^{15}$N measurements. Solubility correction in connection with the CH$_4$ measurements is quite large (~6%) in this method

due to the high ratio of sample/vessel volume which yields a high headspace pressure that contributes to more gas getting

stuck in the refrozen ice (Fegyveresi, 2015). The calibrations (volume and temperature) are briefed in the supplementary

information. Data from PSU-δ$^{15}$N measurements is published here: (Sowers, 2018).

### 3 Cut bubble correction

Air bubbles at the surface of the sample are cut during sample preparation resulting in air loss. Therefore, TAC

measurements need to be corrected for the so called 'cut bubble effect' (CBE). The CBE correction approximates to 10%

near the close off depth and decreases to around 1% in deeper strata (Martinerie et al., 1990). Martinerie et al. (1990)

derived the formula for the cut bubble effect assuming spherical bubbles:

$$TAC = \left(1 - \frac{1}{2}\langle D\rangle\frac{A_S}{V_S}\right)^{-1} \cdot TAC_{raw} \tag{4}$$

Where, D is the average bubble diameter in the sample and $A_S$, $V_S$ are sample surface area and volume, respectively. In

the current study, only samples analysed at PICE had their bubble diameters measured. A photograph of each sample is

taken (supplemental Fig. S1) from which an average of 20 bubble diameters is taken as the sample bubble diameter. The

average bubble diameter of every sample and the corresponding CBE calculations are provided in the supplementary

information. TAC data from PSU are a by-product of methane concentration and δ$^{15}$N measurements. Bubble diameters

have not been measured for these samples. In this study we estimate the CBE for the PSU data from the PICE data.

Through the Holocene section of the RECAP core bubble diameter is decreasing with depth. This is expected as the

bubbles are compressed by the increasing pressure of the overlaying ice. Therefore, down to the YD-Preboreal transition

at 532.6m below surface we calculate the CBE for the PSU data from the linear regression in the PICE data (120-530m).

For samples below 532.6m we use the corresponding average of bubble diameters in the PICE data.

### 4 Results and discussion

The TAC data are presented on the RECAP GICC05 age scale (Simonsen et al., 2018) in Fig. 2. The sample sizes,

extraction devices and measurement procedures are different at PICE and PSU. Correlation plots of the data from PSU

are made to analyse their deviations from the TAC data obtained from the barometric method at PICE (supplementary

information). The data sets show a good correlation with the vacuum volumetric data obtained at PICE. However, on

individual data points differences can be significant resulting from up to one meter depth difference to the closest

correspondent and rapid fluctuations of TAC. The pooled standard deviations of TAC are for PICE, PSU-CH$_4$, and PSU-

δ$^{15}$N are 6.67, 6.80, and 6.11 cm$^3$ kg$^{-1}$ respectively excluding samples with obvious melt features indicating that the

dispersion in the data sets do not differ significantly with the methods of measurement.





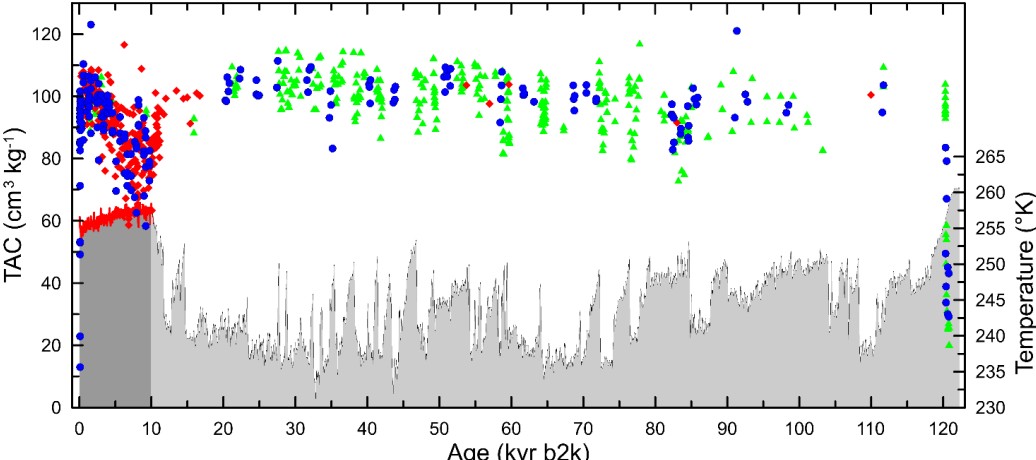

**Figure 2: TAC of the RECAP core on the RECAP GICC05 time scale and estimated mean annual Renland temperature. TAC from PICE, PSU-CH₄, and PSU-δ¹⁵N as blue dots, red diamonds, and green triangles, respectively. Red line Renland Holocene temperature reconstruction (Vinther et al., 2009). Black line NGRIP temperature reconstruction (Kindler et al., 2014) scaled up by 13°K according to the Holocene temperature offset between the two sites. Time scale is b2k (before A.D. 2000)**

### 4.1 Holocene

As outlined in Vinther et al. (Vinther et al., 2009), it is expected that the altitude of the ice sheet was constant over the course of the Holocene. Consequently, TAC will be constant except for changes related to temperature and insolation. The temperature effect will be below 1%. The effect of insolation on the Greenland Holocene TAC has been estimated from data from NEEM, Camp Century, and GRIP (NEEM Community Members, 2013). The increased insolation at the beginning of the Holocene compared to today resulted in a reduction of TAC of about 5 cm$^3$ kg$^{-1}$.

The processes likely responsible for changes in TAC and $O_2/N_2$ ratio are similar in Antarctica and Greenland (Bender, 2002). The accumulation rate determines the exposure time of the surface layers to insolation which may result in more or less sensitivity of the $O_2/N_2$ ratio to insolation (Suwa and Bender, 2008) and also TAC. Given that Renland experiences more than double the accumulation rate than the central Greenland cores, we see the 5 cm$^3$ kg$^{-1}$ change over the Holocene as a maximum. In fact, we suspect that Renland may experience very little insolation driven TAC change.

From present day throughout the Holocene period (0 to 11.7 kyr b2k), the observed TAC values are lower than expected. Especially, during the climatic optimum (6 ka to 9 kyr b2k) the values are as low as ~ 80 cm$^3$ kg$^{-1}$. This change cannot be explained by insolation changes. Line scan can detect melt layers thicker than 2mm and the RECAP line scan record indeed shows numerous melt layers (an example in Fig. 3). However, observations of melt layers decrease with depth because they become too thin to be detected (Taranczewski et al., 2019). Though caution is exercised in avoiding the visible melt layers in our samples, TAC measurements during the Holocene climatic optimum (~501-529m) show significantly low values which is indicative of melt layers untraceable by the line scan. The melt-affected TAC values cannot be used for elevation calculations, but they can still be used for estimating the melt fraction in the core. We will use the Holocene TAC data to derive the melt fraction by assuming a linear relationship between the TAC and the percentage melt in a sample (Herron and Langway Jr, 1987). To construct this linear relationship, we use theoretically calculated TAC for RECAP ice with present day conditions and refrozen water equilibrated with the atmosphere based on Henry's solubility law (supplementary information). Theoretical TAC is used as the measured TAC is inevitably affected by untraceable melt. Still, theoretical calculations estimated melt fractions from selected samples agree reasonably well. We calculate the melt fraction from the insolation corrected TAC (NEEM Community Members, 2013) as well as for the uncorrected TAC data from 100 year averaged TAC data (Fig 4).



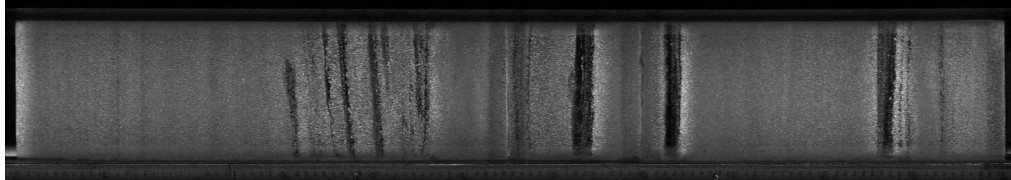

**Figure 3: Line scan image of Bag 143 and 144 of the RECAP core showing melt layers**

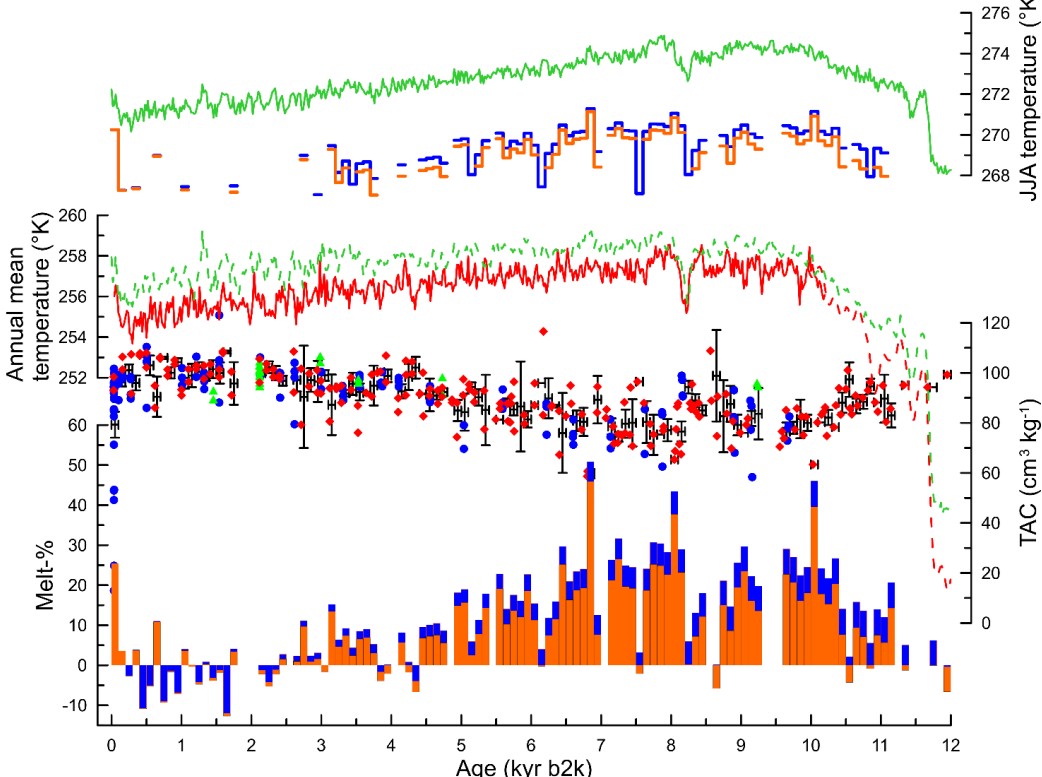

215

**Figure 4: RECAP Holocene TAC with corresponding melt-%, and derived summer temperature together with other temperature estimates. TAC from PICE, PSU-CH4, and PSU-δ15N as blue dots, red diamonds, and green triangles, respectively with 100-year averages and 1 sigma standard error. Blue and orange bar charts are melt-% calculated from the mean TAC values excluding (blue) and including (orange) a correction for insolation changes. Blue and orange step plots are temperature reconstructions from the melt-% again with (orange) and without (blue) taking an effect from insolation changes into account. Red line: Renland annual mean temperature reconstruction (Vinther et al., 2009). Red dashed line: temperature from NGRIP increased by 13° (Kindler et al., 2014). Green lines are annual mean (dashed) and JJA (solid) temperature calculations for Renland (Buizert et al., 2018).**

### 4.1.1 Holocene summer temperatures inferred from melt fraction

225    We now use the estimated melt fraction to infer local summer temperature at Renland. Langen et al. (2017) extended the subsurface scheme of the HIRHAM5 regional climate model to include snow densification, varying hydraulic conductivity, irreducible water saturation and other effects on snow liquid water percolation and retention calculate melt water production. It allows us to derive an empirical relationship between melt fraction and temperature in the region (Appendix A). Applying this relationship to the Renland melt fraction, we derive past summer surface temperatures (Fig.

230    4).



The extrapolated temperatures from melt fractions suggest that Holocene summer temperatures in Renland were ~2 to 3°C warmer than the present day (Fig. 4). This is consistent with temperature reconstructions from δ$^{18}$O signals of Agassiz and Renland which reveal that, during the Holocene climatic optimum, Greenland temperatures were higher than the present day by ~2°C (Vinther et al., 2009). GRIP paleo temperatures interpreted from the δ$^{18}$O profile and borehole temperature measurements also reveal that Greenland was warmer in the climatic optimum (8 kyr-10 kyr BP, boreal) by ~ 3 to 4°C (Johnsen et al., 1995; Dahl-Jensen et al., 1998). We note that melt layers are basically missing in the last 2kyr, increasing only in the last century. This is in line with the observations from Taranczewski et al. (2019) based on line scan images.

**4.2 Previous interglacial (Eemian)**

Greenland surface temperatures were warmer during the Eemian period than in the Holocene. At NEEM, it is estimated that at 126 kyr B2k, the temperature peaked at 8±4 degrees Celsius above the mean of the past millennium (NEEM Community Members, 2013). GISP 2 δ$^{18}$O records also indicate temperatures ~4 to 8°C warmer than the present around 126-128 kyr BP (Yau et al., 2016).

The TAC signal of RECAP in the Eemian section (> 119 kyr b2k) has incongruously low values (as low as ~ 20 cm$^3$ kg$^{-1}$, Fig. 5). It is likely that this is due to melt occurring due to increased temperatures. Applying the same metric as for the Holocene, the observed low TAC originates from temperatures at least 5°C warmer than today. This estimate disregards insolation changes which are comparable to today 120 kyr ago. Higher temperature results in higher pore volume (Martinerie et al., 1994) resulting in higher TAC. For each degree increase, the pore volume becomes half a percent larger. Therefore, we see our estimate of 5°C warmer as a minimum. This assumes that none of the TAC changes are caused by elevation changes. If the higher temperature has led to a decrease of the Renland ice cap, TAC has to increase, again making the estimated 5°C temperature change a minimum.

Generally, melt layers lead to spikes in the CH$_4$ record due to the higher solubility of methane compared to bulk air or in situ production (see e.g. NEEM Community Members, 2013). TAC from the NEEM ice core in the Eemian interglacial shows low TAC values, with spikes in the CH$_4$ and N$_2$O records which is a clear indication of the presence of surface melt (NEEM Community Members, 2013). Surprisingly, we do not see spikes in the on-line CH$_4$ record of RECAP in the Eemian section (Fig. 5). The low gas content of the ice core in combination with the extremely low depth resolution (554-562 m corresponds to 119 to 120.8 kyr b2k) smoothed the CH$_4$ record. Potentially the melt spikes are just not visible any longer.

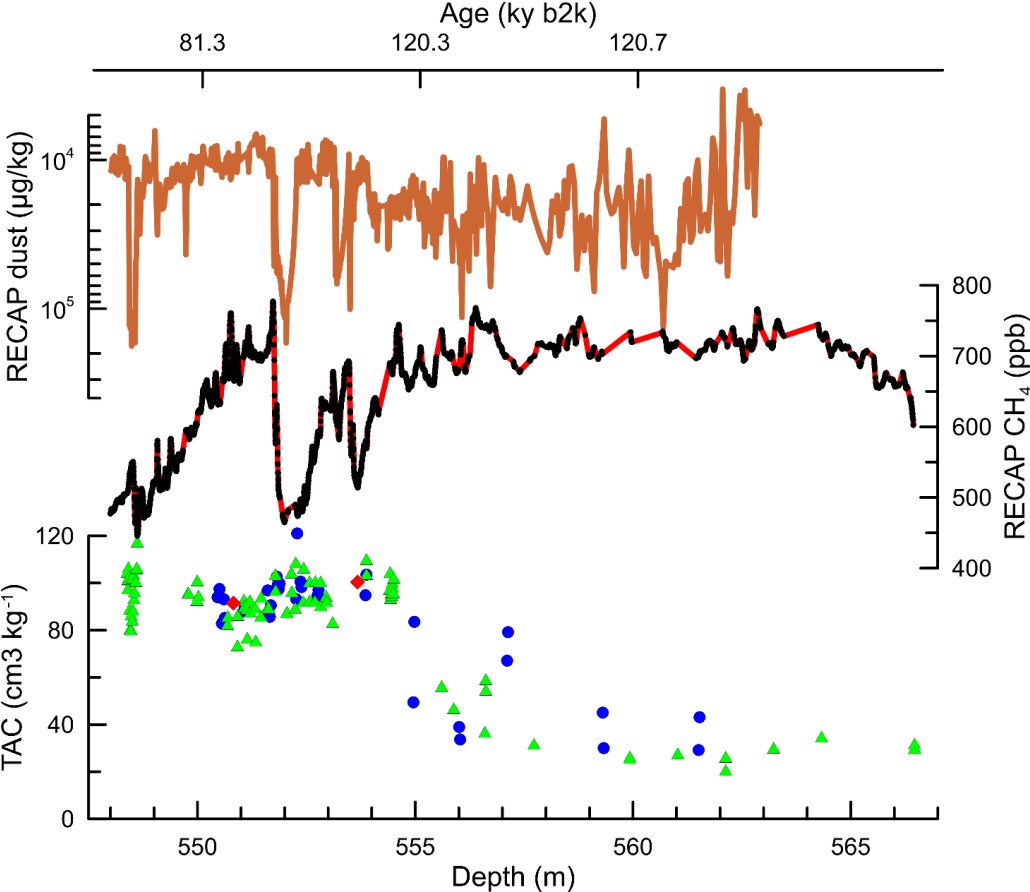

**Figure 5: Recap section showing low gas content, starting around 555 m below surface. TAC from PICE, PSU-CH₄, and PSU-δ¹⁵N as blue dots, red diamonds, and green triangles, respectively. Red line RECAP dust record (Simonsen et al., 2018) and on-line CH₄ measurements.**

### 4.1 Glacial

The TAC of RECAP in the glacial section (11.7 to 119 kyr b2k) shows overall similar values as at present day (Fig. 2). However, like for NGRIP (Eicher et al., 2016), we find TAC variations associated with D-O events that are not related to elevation.

### 4.3 Relation to climate changes during D-O events

Generally, in the vicinity of the D-O events, the RECAP TAC signal is dropping rapidly, recovering after a few hundred years. The variations we see are on the order of 10-20%. If those changes were related to elevation changes, they would correspond to several hundred meter changes, which is unrealistic in only a couple of hundred years. It is more likely that the changes are related to changes in pore volume. Similar effects have been observed in the NGRIP core (Eicher et al., 2016). An increase in temperature will with constant pore volume result in a small reduction of TAC. This effect is slow to take effect because changes in surface temperature must first reach the close-off depth through thermal diffusion. Once steady state is reached, the effect is counterbalanced by a slightly bigger pore volume (Martinerie et al., 1994). As for the NGRIP site (Eicher et al., 2016) we dismiss synoptic pressure changes as a primary cause for the observed changes in TAC. To analyse the effect further and compare to the NGRIP site, we took the following approach. Dynamical effects in TAC can be expected from the moment of change till a new steady state is established. For the firn column this is when



e.g. at a D-O event the higher accumulation snow has reached close-off. As methane and temperature changes have been found to occur in close temporal proximity, the depth interval to be considered for a dynamical change is between the depth when methane changes are observed and the depth where changes in parameters recorded in the ice occur, e.g. $\delta^{18}O$ of $H_2O$ or dust. For the RECAP ice core, we find that this depth interval corresponding to $\Delta$age is quite variable. Why this is the case we ignore. We choose to produce a stacked plot over D-O events with normalized time axis. For each event the time axis is normalized so that the methane transition (in some events defined by change in $\delta^{15}N$) is set to 1 and the decrease in dust (coincident with the change in $\delta^{18}O$) is set to 0. We treat the Eicher et al. (Eicher et al., 2016) dataset for NGRIP in a similar way. The detailed results of this approach for RECAP and NGRIP can be found in the supplemental plots S8. The results for TAC are shown in Fig. 6 as a lowpass cubic spline fit with a 200-year cutoff period, according to Enting (Enting, 1987) with 1 sigma uncertainties for the spline fit. The uncertainty is obtained by randomly varying the data points within their error before calculating 1000 Monte Carlo splines.

For both cores, on average, the TAC values start to decrease around the depth (time) when $CH_4$ starts to increase at the beginning of a D-O event. However, the minimum TAC is found before the depth (time) when D-O manifest as drop in dust or increase in $\delta^{18}O$. For NGRIP this minimum is reached 600 years before the snow associated with the D-O event reaches close off while for RECAP it is about 100 years. Also, the drop in TAC is more significant for RECAP than for NGRIP. Overall, TAC variations associated with D-O events as recorded in RECAP are ~30% higher than NGRIP. This may be a result of the higher accumulation and temperature at Renland.

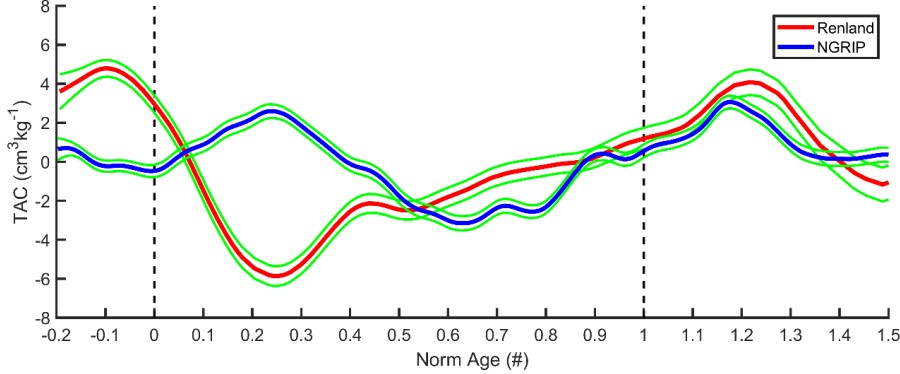

**Figure 6: Effect of D-O events on TAC signals for RECAP (red) and NGRIP (blue) on a normalized age/depth scale over the past firn column. Surface is at depth 0 and close off is at depth 1. Presented is the Enting spline (Enting, 1987) with a 200 year cut off period with green lines giving the one sigma uncertainty of the spline.**

### 4.3 TAC and insolation

It has been observed that the influence of insolation on the firn structure at Antarctic sites is profound (e.g. Lipenkov et al., 2011; Bender, 2002). The TAC records from Antarctic ice cores show a pronounced correlation with integrated summer insolation (ISI) as seen in EPICA DC (Raynaud et al., 2007) where the correlation is 0.86 ($r^2$), whereas the correlation of the NGRIP glacial TAC signal is only $r^2 = 0.3$ (Eicher et al., 2016).

For the RECAP core, the correlation ($r^2$) of spline of ISI (sum of annual insolation $\geq 380W/m^2$) filtered with a cut off period (COP) of 3000 years (Enting, 1987) and the low pass spline of glacial TAC (11.7 to 119 kyr b2k) filtered with COP of 750 years is obtained as 0.004 (Fig. S9). As lined out earlier, we may see a pattern where higher accumulation rate, due to reduced exposure time of the firn structure, results in reduced influence of insolation on TAC. However, we do not rule out that the effect may be masked in the RECAP core by the high variability associated with D-O events. A higher sample resolution allowing to exclude data affected by rapid climate change from the analysis could clarify how accumulation rate and insolation interact.



### 4.4 Elevation calculation for the last glacial maximum

We finally make an attempt to use glacial TAC to reconstruct ice sheet elevation. To do so we need to avoid periods of rapid climate changes. Only the last glacial maximum section fulfils that criterion and has good data coverage (Fig. S10).

For a meaningful interpretation of past elevation changes, TAC data generally need to be corrected for upstream flow, summer insolation influences, surface melting and effects of temporal variations. Since the RECAP ice core is drilled near the dome of the ice cap, upstream correction is not necessary. The melt affected TAC data in the Holocene and Eemian sections are not used for elevation calculations. No melting is expected in the glacial section.

As discussed in the previous section, since the RECAP TAC has negligible correlation with ISI, insolation correction

may be unnecessary. We calculate elevation with and without accounting for insolation changes where we apply the TAC correction according to (NEEM Community Members, 2013). From TAC the local ambient pressure ($P_c$) can be estimated where we need to estimate local temperature ($T_c$) and pore volume ($V_c$) applying Eq. 1. We estimate the past local temperature from NGRIP (Kindler et al., 2014) where the NGRIP record is increased by 13°C according to the present day difference between the NGRIP and RECAP sites. The average pressure of the 21 samples in the LGM period comes

to 744 ± 5 mbar (1 standard error). The insolation correction for this period is -10 mbar. Uncertainty of $T_c$ and $V_c$ are significant. Each centigrade changes $P_c$ by 4 mbar and 1% change in $V_c$ results in a 7 mbar change in $P_c$.

The pressure $P_c$ can now be interpreted in terms of elevation based on the barometric formula, Eq. (2). Unknowns are the past near surface lapse rate, and the pressure at sea level.

Along the Greenland ice sheet, the annual mean near surface lapse rate (dT/dz) has been calculated as -7.1 °C km⁻¹ based

on the data obtained from the 18 automatic weather stations for the period 1995-1999 (Steffen and Box, 2001). The lapse rate varies largely over the year from -4 °C km⁻¹ in summer to -10 °C km⁻¹ in winter (Steffen and Box, 2001). However, for present day Renland we calculate a near surface lapse rate of -4.5 °C km⁻¹ where our point of reference is Illoqqortoormiut about 200km from RECAP with $T_a$ of 265.5 °K, $h_a$=0 m, $P_a$ =1012.2 mbar (Cappelen et al., 2001).

Our calculations are relative to the present sea level. Krinner et al. (2000) suggest that sea level pressure at current sea

level was slightly higher than today during the LGM. From Fig. 2 in Krinner et al. (2000) this increase is between 0 and 5 mbar. In the following we disregard the uncertainty but increase the past sea level pressure to 1015 mbar. A model study on the LGM lapse rate concludes that the LGM was about 2°C km⁻¹ lower than today (Erokhina et al., 2017). Based on the present day lapse rate we calculated a LGM lapse rate of -6.5°C km⁻¹. One could argue that the observed lapse rate for Renland of today is above the observation for Greenland since we measure the temperature in the RECAP firn. We

do know that there is melting occurring today and we therefore may underestimate the annual mean temperature at Renland. Therefore, we also calculate with the lower lapse rate of -9.1 °C km⁻¹ (again lowered by 2°C from modern).

Due to the topography of the ice sheet, the expectation is that the Renland ice sheet elevation is similar to today at 2340 m above present see level. Without insolation correction we calculate 2259 m and 2286 m for lapse rates of -7.1°C km⁻¹ and -9.1°C km⁻¹, respectively. Including insolation correction, the numbers climb to 2354 m, and 2384 m for the two cases.

The statistical uncertainty is ± 50 m (1 standard error) which does not include any uncertainty on $V_c$ or $T_c$. E.g. including a 2°C uncertainty in $T_c$ combined with a 2% uncertainty in $V_c$ increases the uncertainty of the calculations to ± 220 m, enough that any of the four calculations covers the assumed ice sheet elevation of 2340 m above present see level for the LGM.

### 5 Conclusion

We measured TAC back to 121 kyr b2k from the Greenland RECAP ice core. The TAC signal has unexpectedly low values in the early Holocene (6 to 9 kyr b2k) and during the Eemian (119 kyr to 121 kyr b2k). The low TAC values in the Holocene period point to melt events as corroborated by elevated CH₄ values in the RECAP core (Vladimirova,



personal communication 2019). Melt fractions calculated from the RECAP TAC signal in the Holocene are in turn used to interpolate the summer surface temperatures (subsurface HIRHAM5 model). Summer temperatures in early Holocene at Renland were ~2° to 3°C warmer compared to today. This finding is in agreement with similar findings from Greenland ice cores and model calculations. During the previous interglacial we see significant melting that let us conclude that temperatures at Renland were at least 5°C warmer than today.

The influence of local summer insolation ≥ 380 W/m$^2$ on the TAC signal of Renland is minimal as indicated by the correlation coefficient (r$^2$) of 0.004. Elevation of the Renland ice cap is calculated from the last glacial maximum TAC data. These elevation calculations encompass the uncertainties that arise from the assumption of the lapse rate, temperature and pressure gradients that existed in the past and show that within uncertainty the elevation has been similar to today. During D-O events, RECAP TAC shows significant variations that are larger than in other ice cores. How these variations come about is currently not understood. The stacked data analysis that we performed for RECAP and NGRIP show that changes in the firn structure must occur within the firn column during events of rapid climate change.

**Appendix A:** We use the estimated melt fraction to infer local summer temperature at Renland. Langen et al. (2017) extended the subsurface scheme of the HIRHAM5 regional climate model to include snow densification, varying hydraulic conductivity, irreducible water saturation and other effects on snow liquid water percolation and retention calculate melt water production. It allows us to derive an empirical relationship between melt fraction and temperature in the region (Appendix A). The model takes weather forcing at the surface from the regional climate model HIRHAM5 over the period 1980-2016 (forced in turn by ERA-Interim on the lateral boundaries) and calculates surface melting (of snow and bare ice), vertical percolation, retention, refreezing, densification, grain growth, runoff and surface mass balance. The subsurface calculations are performed on the 5.5x5.5 km grid of HIRHAM5. Over 15 grid cells centred on the RECAP drill site (Fig. 5), we gather total annual meltwater production and JJA average temperatures from each of the years 1980-2016 (giving 15 x 37=555 data points).

Meltwater production is converted into melt percentages using an observed approximate annual accumulation rate of 50 cm ice equivalent. The data points are then divided into 0.5 K bins with respect to JJA temperatures. For each bin, a mean melt percentage is calculated (Fig. A1). An exponential fit describes the resulting relation between JJA temperatures and melt percentages as: %-melt = b*exp(a·T), b= 1.3145·10$^{-76}$, a = 0.6585

where T is the mean JJA surface temperature in K.

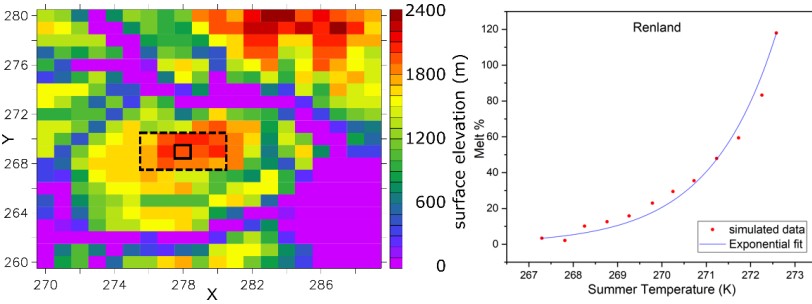

**Figure A1: left) 15 grid cells (dashed square) centered on the RECAP drill site (solid square) on the subsurface calculations performed on the 5.5x5.5 km grid of HIRHAM5. right) Renland surface temperatures based on melt fractions**

**Acknowledgements:** The RECAP ice coring effort was financed by the Danish Research Council through a Sapere Aude grant, the NSF through the Division of Polar Programs, the Alfred Wegener Institute, and the European Research Council under the European Community's Seventh Framework Program (FP7/2007-2013) / ERC grant agreement 610055 through



the Ice2Ice project. The Centre for Ice and Climate is funded by the Danish National Research Foundation. PLL gratefully
acknowledges the contributions of Aarhus University Interdisciplinary Centre for Climate Change (iClimate, Aarhus
University). Thomas Blunier acknowledges support from the Carlsberg Foundation and Australian Antarctic Program
Partnership. We thank Todd Sowers for measurements at PSU and for decades of fruitful collaborations.

**Author contributions.**

*Sindhu Vudayagiri:* TAC measurements at PICE, data collection, data analysis and drafted the manuscript.

*Johannes Freitag:* Line scan images.

*Peter L. Langen:* Simulated JJA surface temperatures based on the meltwater production (subsurface model calculations
performed on the 5.5x5.5 km grid of HIRHAM5) in Renland.

*Bo Vinther:* Contributed to data analysis and manuscript preparation.

*Thomas Blunier:* Designed the experiments, made the final data analysis, and final manuscript preparation.


**Data availability.** Data not corrected for CBE from PSU-$\delta^{15}$N measurements can be found here (Sowers, 2018). The full
dataset is available at the Arctic Data Center (Blunier, 2024).

**Competing interests.** The contact author has declared that none of the authors has any competing interests.

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
