# Peer review of "Total Air Content measurements from the RECAP ice core"

_EGUsphere, 2024_

## Author Comment (AC2)

Total Air Content measurements from the RECAP ice core
Sindhu Vudayagiri, Bo Vinther, Johannes Freitag, Peter L. Langen, and Thomas Blunier
**Handling editor**: Amaelle Landais, amaelle.landais@lsce.ipsl.fr

**Our point-by-point response to the two reviewers' comments below.**

**RC1**

**Referee comment for: 'Total Air Content measurements from the RECAP ice core', Vudayagiri et al.**

This manuscript presents Total Air Content (TAC) measurements throughout the full RECAP ice core from the Renland ice cap, Greenland. The initial aim was to understand the elevation history of the ice cap, as TAC of ice core bubbles can be a proxy for ice elevation given the relationship between elevation and barometric pressure. However, much of the measured record was affected by either melt induced lowering of air content values, or rapid climatic transitions dominating the TAC signal due to non-elevation induced effects on pore volume. The main interpretations therefore presented are a melt correction for the TAC in the Holocene section of the core, and an elevation calculation for a short section of the core in the Last Glacial Maximum which is presumed unaffected by melt or strong climatic changes.

Much analysis has been conducted in this work, and it would be of benefit to see these results published. It is nice to see TAC measurements from two laboratories each with established and robust methodologies, which can demonstrate analytical comparability, especially given the more complex outcomes of the data. The authors have commendably looked for additional interpretation of the data given that much of it could not be used for elevation.

For me, the structure and writing of the paper lacked clarity, making it difficult to follow. The rational of the paper seemed somewhat unclear – the $d^{18}O$ record suggested as an anchor for the hypothesis of no elevation change at Renland isn't presented or discussed further, and previous studies of Greenland TAC (NGRIP) have shown climatic influences affecting TAC values so I am not clear on how an elevation history was expected to be obtained throughout the whole of RECAP. I am also missing what the wider scientific conclusions of the paper are, for example could there be some discussion on the application of TAC measurements to Greenland cores given the observations in this paper? Everything needs pulling together more succinctly for the reader. In some areas, I would like to see further detail, for example with the age scale and with errors on the different analytical systems used. I give more specific comments below.

We fully agree with the reviewer, that most of the manuscript is discussing obstacles for interpreting TAC in terms of elevation. Only during the LGM do we believe the RECAP data to allow such calculation. We see that we need to make this clearer in the abstract and the conclusions. We see this manuscript as a step towards more research for understanding TAC, which we believe to be an important parameter in the climate puzzle also for the near future.

There is a misunderstanding concerning the d18O of H2O data. Data from Renland (older Renland core from 1988) has been used in Vinther et al., 2009. It is not presented here. We will make this clear in a revised version.

**Major questions:**

Abstract:

The abstract somewhat peters out, not effectively summarising the paper, and I suggest it is rewritten. Given the main goal of the paper was for an elevation study, why does it not give the elevation estimate for the one section of the core which allowed it? The final two sentences state two things that were done in the paper without presenting the result of them. There are no final remarks on the scientific takeaways of this paper. Improving this may help the clarity of the rest of the paper.

We fully agree and will modify the abstract accordingly.

Introduction:

Line 28 states that 'the study uses $d^{18}O$ of $H_2O$ from the Renland ice cap as an anchor point arguing that the ice cap has not experience significant elevation changes'. However, I could not see this record referenced here or presented in the paper. This should be included. A recent paper by Grieman et al., 2024 present an example of the use of $d^{18}O$ in conjunction with TAC for elevation change, published I note after submission of this paper but a useful reference for the revision. (https://doi.org/10.1038/s41561-024-01375-8).

The statement refers to the study of Vinther et al., 2009. We will make that clear in a revised version of the manuscript. The Grieman study refers to a different situation and approach and we see no need to cite it here.

Section 1.2:

This section would benefit from subheadings to break the text in to defined sections. The discussion on insolation is long for a general overview of the idea; can this be refined? I also think this discussion needs to be connected back to the expected or observed influence of insolation on the RECAP core itself to keep it relevant.

We do agree that the introduction needs some clarifications (also reviewer 2). We will improve the structure adding a subheading as suggested. We will also shorten the lengthy historical introduction to the development of the dependence of TAC to insolation as suggested. As the effect is discussed later in the manuscript with respect to what we observe in Renland some introduction is necessary in this introductory part.

Line 105: It is referenced that the TAC record in the NGRIP core was influenced by D-O events and the conclusion was that climatically stable periods are preferential for elevation estimates. Therefore, why was the rational of the TAC measurements from RECAP to show elevation history of Renland throughout the core, when it would also likely show these same climatic influences over significant periods? Was it expected that you would observe something different?

When we started the study, we were not aware of the complications and the goal has been to confirm that the elevation was stable from the last glacial to today. Especially the short term millennial fluctuations came as a complete surprise to us.

1. Measurements:

TAC from PSU was measured as part of $d^{15}N$ analyses. Given that $d^{15}N$ can be a useful indicator for firn column changes, could this available $d^{15}N$ be helpful in the interpretation? It looks as though lower TAC values coincide with higher $d^{15}N$ in the Supplementary Information plots. The plotted $d^{15}N$ is not mentioned in the main paper, only a data repository link is given – can the main paper refer to these plots, or even include the $d^{15}N$ in a plot in the main paper? I think there is interest here.

The main influence on d15N over D-O events is thermal fractionation. It is useful to determine when transitions start. We can mention d15N in the main manuscript.

What are the errors of the respective analytical systems used at PSU and PICE? I can see final pooled standard deviations of the data but not the errors of the analytical systems themselves.

The analytical errors are below the natural variability of TAC.

2. Results and Discussion:

The TAC data are reported to be presented on the RECAP GICC05 age scale, but the reference for this seems to be incorrect; the Simonsen 2018 paper does not appear to discuss an age scale. Please replace with appropriate citation or included some detail on the age scale in this paper.

The reviewer is correct, this is the wrong reference. Will be changed to Simonsen et al., 2019 also in the figure 5 caption and the supplemental material. Thank you for spotting this mistake.

Is the existing RECAP GICC05 age scale an ice age scale only, or did it include an existing gas age scale? I am not clear on whether this paper has constructed their own gas age scale. If the latter, can the gas age scale please be presented somewhere? For clarity, it would also be helpful if it could be stated somewhere which data are presented on which of these two age scales.

We use the time scales published in Simonsen et al., 2019. For most plots the difference between ice age and gas age is not visible. We will add information on the time scale used to all figures.

3. Conclusions:

I would like to see the concluding section bring together the investigations of the paper more effectively. I am missing final remarks on the wider scientific impact of this paper – for example given the results of this paper, what are the suggestions for future TAC analysis in other Greenland cores?

We agree and will do so.

**Minor/technical points:**

Abstract: Suggest changing TAC 'is' a proxy to TAC 'can be' a proxy, in the complications of the proxy and the context of the results of this paper.

Ok

Line 23: Can you say why Renland ice cap elevation stability is a pertinent question?

Generally, this not questioned but we believe it is better to confirm that assumption.

Line 39: at an elevation 'of' 2315 m a.s.l. … near 'the' summit.

Fine

Line 54: total air content has been defined already so can just be TAC, also missing a comma after TAC.

OK

Line 71: Can you clarify whether the parameter you use is the one with or the one without the later additional sites?

It is already stated that we use the one without the additional sites in line 70.

Line 105: This it the first mention of Dansgaard-Oeschger events in the main text I think, define 'D-O' here since it is used herein.

OK

Line 189: 'The temperature effect will be below 1%' – can you explain or reference this value?

This results from the ideal gas law and we believe it does not need further explanation.

Line 191: (in) TAC

We do not understand what the issue is here, sorry.

Line 199: Line scan(ning)

Line scan is a term that is understood and used in the community.

Line 203: Line scan(ner)

Line scan is a term that is understood and used in the community.

Figure 4: The top step-plot of melt % does not appear to have an axis. Please amend this.

We will make clear which axis applies to the plot.

Lines 225 – 228: This has been directly copied and pasted from the Appendix which it refers to (or vice versa, since I notice the same section in Appendix A still self-refers to Appendix A. Please reword one to avoid repetition and remove self-reference in Appendix.

We will do so.

Numbering of subsections is incorrect after 4.2.

Thank you for noticing.

Line 279-286: Again this section is directly copy and pasted from/to the Supplement so needs re-writing. In both sections, 'Why this is the case we ignore' seems quite a negative phrasing to me, perhaps it just a preference but I would ask to change this. You don't explain this further because you don't believe it impacts your results, or because you do not know the explanation?

We do not have an explanation. It is very strange compared to other cores.

Line 307: 'outlined' rather than 'lined out'

OK

Supplement:

The supplement needs a thorough proofread by the authors for the present grammar/typing errors.

We agree.

S1: Is the bubble size determined by eye looking at the photos with the scale? If so, does this have quite a large error?

The bubble diameters are measured with a calliper. The uncertainty of the bubble diameter is substantial by natural variations and the measurement uncertainty. Figure S3 shows standard errors for the bubble diameters on bag means (55cm sections).

Figure S8: Difficult to read this figure at the size presented.

We will increase the size of the figure.

'DO-events' is used in the supplement, while it is 'D-O events' in the main text. Should be the same for consistency.

Will be fixed.

**RC2**

The manuscript by Vudayagiri et al. present a new Total Air Content (TAC) record measured on the RECAP ice core from Greenland and covering the past ~121 000 years. The authors combine TAC dataset measured in two laboratories. The first dataset has been measured using a new experimental set-up dedicated to TAC analyses at PICE, Niels Bohr Institute (Denmark) and the second dataset has been obtained indirectly using the existing experimental set-ups for $CH_4$ and $\delta^{15}N$ of $N_2$ analyses at Penn State University (USA). The authors show that it is challenging to use the new TAC record to infer robust information about past elevation changes at RECAP. Still they make an attempt for the last Glacial Maximum and they follow other research directions to extract climatic information from their new record (e.g. melt fraction and amplitude of the warming during the mid-Holocene and at the end of the Last Interglacial) and investigate the controlling factors of both the short- and long- timescale variations in the RECAP TAC record (e.g. firn structure changes, local insolation).

I'm surprised (to say the least) about the shape of this manuscript especially considering the list of experts who worked on it. Indeed, the current manuscript requires some very major revisions both in terms of the form and the content in order to reach a state that is reasonable enough so it can be considered for publication. In fact, I think that it is a real pity that this manuscript is in such poor shape because it is nonetheless very nice to see a new TAC record measured at high resolution on a costal site in Greenland. I think it is important that this new record is eventually published. I also think that the different research directions based on this new records which are presented (but never really fully developed) are interesting and well within the scope of Climate of the Past.

We would like to thank the reviewer for their effort and their extensive and very detailed review. They are obviously knowledgeable about the topic, and we value their time and effort to work through our manuscript which, we know, is not an easy read due to the complexity of TAC.

We agree that recent findings, including the ones presented in this manuscript, show that TAC is quite difficult to understand. There are a number of pitfalls, and we can make that more clear in the conclusions and abstract as requested (also by reviewer one).

**Major comments:**

Overall, I found the current manuscript very difficult to read and to understand. I would argue that it is due to the fact that the manuscript contains a number of big problems both related to the form and the content. Indeed:

- o   I find the form and structure of the manuscript of poor quality. It is written as such that the logic followed for the development of the different paragraphs and the different sections is often hard to grasp, it often lacks of transitions between the paragraphs. In addition, the captions of the figures are not precise enough and the figures themselves often not big enough making hard to see what the authors are describing. I also believe that there are too much important information and calculations, and too many figures that have been placed in the supplementary material (SM), making the reader having to go constantly back and forth between the main manuscript and the SM, this is not ideal.

- We will do our best to make the text easier to follow. The issues with TAC in the RECAP core are clearly different between the Holocene, the glacial and the Eemian section. The structure of the manuscript reflects that. This naturally leads to clear breaks. We believe that the nitty gritty (but important) details are all put to the supplementals where they belong allowing the reader to get the main message of the manuscript from the main text without distraction. From following up remarks it seems that what the reviewer is missing in the figure caption is the information about the time scale used (ice age or gas age). The difference is not visible but we will specify which age scale is used.

  - In many places, the work performed is either not well described and vague (in several places, background information is missing), or in opposite, sometimes, the information becomes very technical and no background information is provided to help the reader to follow, especially in the case that the reader is not a specialist of this climate proxy. All this makes it very hard to identify properly what the key results coming out of this study are.

  - We happily add the specific information requested listed under minor comments.

  - Various research topics related to TAC are being tackled in the manuscript going from investigating the melt fraction based on TAC to reconstruct summer temperature, to using TAC during the Last Glacial Maximum (LGM) to provide information on paleo-elevations, to investigating the link of TAC changes at RECAP with insolation changes, and also how they are affected by the abrupt Dansgaard-Oeschger (DO) events. All these topics are worth looking into but maybe it is too much for one paper? If the authors want indeed to tackle them all, they need to do an important effort to guide thoroughly the readers by providing a more comprehensive introduction providing referenced background context, presenting in both a succinct and precise way the different aspects, announcing the outline of the different following sections etc. Alternatively, the authors could consider refocus their study on only a selection of topics. It could be beneficial to improve the clarity and structure of the paper and it would also prevent having such a large SM.

  - We do not believe in minimal publishable units and prefer to publish all TAC results of the RECAP ice core in one manuscript. We believe that this is the best way to communicate the complexity of TAC in this core to the reader.

  - In turn, I would like to mention that as far as I understand this is the first time that the experimental system developed at PICE is presented in a publication. As a result, I believe that stronger emphasise should be put on the presentation and the assessment of the new system e.g. no clear description and quantitative estimates of the attached uncertainties is provided. To me, the description of this new experimental system is an important result of the study in itself and it would deserve also to be highlighted, for instance in the abstract and the conclusions of the study.

  - We will add a sketch of the system to the supplementals. The system by itself is not special and based on previously published principles. Therefore, we do not see the need for further elaboration on the measurement system. We will, however, include a sketch of the system in a revised version of the manuscript.

In the following I provide more detailed comments information related to these main remarks and suggestions for changes. I would strongly recommend the authors to consider them when preparing a new version of their manuscript.

**-Abstract:**

It needs to be rewritten in order to focus precisely on what is discussed/presented in the paper. Most sentences are very vague. The authors tell us what they investigated but not the actual results found.  Here they should highlight for instance what is new compared to other TAC records, what are the quantitative results related to summer temperature reconstructions. Also, if the authors want to mention the paleoelevations estimates for the LGM, they need to also be more specific and provide the actual calculated estimates. It seems to me that the fact that a new experimental set-up for measuring TAC is important and should be also included in the main messages of this abstract.

Will be done.

**-Section 1.2, paragraph starting line 74:**

This section needs to be revised to follow a linear development as it is very hard to read. It currently jumps from mentioning the short-term variability of Vc to the role of solar insolation on $O_2/N_2$ and then the link between local insolation and TAC. This makes it very unclear and it lakes transitions between the paragraphs. A suggestion would be to (1) introduce the fact that there is a link on long timescales between local insolation and TAC, (2) present the parallel between TAC and the $O_2/N_2$ ratio and the potential physical mechanisms to explain the link of these tracers to local insolation, (3) next, to mention that based TAC has been used for constraining through orbital dating the latest Antarctic reference chronologies (AICC2012 and AICC2023, Bazin et al. 2012; Bouchet et al. 2023), and (4) then present and discuss the fact that TAC is also influenced by short-term climate variations. The authors cite the Eicher et al. (2016) paper but there is also the recent paper by Epifanio et al. (2023).

We principally agree (also see reviewer 1) and will shorten and restructure that section. Following reviewer 1, we will introduce a subheading for perennial variations of TAC. Reference to Epifanio will be added here as well.

**Section 2:**

As mentioned previously and as far as I understand this is the first time that the experimental system developed at PICE is presented in a publication. As a result, I believe that stronger emphasise should be put on the presentation of the system, e.g. it is necessary to add a schematic of the experimental system. Also, it is essential to provide a more substantial description and quantifications of the different sources of uncertainties attached to the measurements and to state the overall uncertainty attached to each analysis of TAC with the new system, e.g. the authors mention briefly the uncertainty related to the calibration of the experimental volume but there are other sources such as the one related to the water vapour pressure evaluation, the mass loss during the time of sample pre-evacuation or the uncertainty related to the measurement of the pressure gauge itself ?

The system follows the well explained principles in Lipenkov et al., 1995. A schematic will be added to the supplementals.

As explained, water vapor is removed in the measurement system, therefore no correction is needed. During the 30 minute evacuation of the system, sublimation occurs, leading to a systematic offset of all TAC measurements. Lipenkov, et al., 1995 estimate the mass loss to 1% in their original system. We do not take that into account.

I understand that the PSU experimental protocol has been presented in details elsewhere, however the authors should still state the uncertainty attached to the TAC measurements performed with this system.

As explained, the PSU system has not been built for TAC measurements. Therefore, we carefully compare their result to the PICE data in the supplementals. No systematic offset is found between the datasets.

In the section 4, the authors provide a pooled standard deviation but (1) I don't think this is the place to provide this information (this should be done in section 2) and (2) I don't understand what it represents effectively and what it does and does not account for. This needs to be clarified.

Section 4 is where the data is presented. We believe this is the place to show that the different datasets are compatible with no offset and similar dispersion. Obviously, the pooled standard deviations calculated are on the final data with all corrections explained earlier. We see no need for clarifications.

Finally, in my opinion, the presentation of this new system at PICE is an important added value to the paper and hence this could be better highlighted for instance in the abstract and the conclusion of the paper.

Happy that the reviewer sees our system as an "important added value". However, we do not believe it that special that it needed more highlights. We will, however, provide a sketch of the system in the supplements.

**Section 3:**

The dataset from PSU are corrected for the CBE using a linear regression defined based on the PICE data, this regression needs to be presented more clearly and discussed. I would suspect the depth-interval considered is going to have an impact on the slope of the regression line and hence the values of the CBE corrections for PSU measurements. Figure S3 presents some error interval but no details are provided regarding how this was quantified, this is missing and should be added.

We will add reference to the supplements in the main text and elaborate on the error estimate presented in S3.

The authors discuss in the supplementary material the agreement between the PICE and PSU dataset. I think this is a shame that it is placed there as this section is very important for the readers to gauge how far it is possible to go into the interpretation of the new records when combined. Also the current discussion is very brief and qualitative, it needs to be revised and more quantitative.

We believe this technical aspect of the datasets belongs into the supplementals. It is referenced in the main text with the main results, namely that the datasets are compatible.

**Section 4:**

The structure of this section needs to be revised. In particular, before going into the interpretation of the different sections of the record, it is essential that clear presentation and description of the dataset are provided e.g. in term of the amplitude of the signals, their variability and how they compare. And then, a structure of the rest of the section needs to be announced so the reader knows what to expect in the following of the section. The titles of the sub-sections 4.1, 4.2 and 4.3 should also be more specific than just the name of a time period. Also, please check the numbering of the sections, there are some mistakes. For instance, a second section 4.1 (Glacial) comes after section 4.2 (previous Interglacial). That section 4.1 Glacial seems too short to be a section of its own too?

Renumbering will be done.

The section related to the identification of the melt layer needs to be re-written so it is easier to follow the reasoning of the authors. They also need to provide some background regarding the identification of the meltlayers and how it is possible to infer some temperature reconstructions from the melt fraction. In fact, how is the melt fraction estimated, what is based on theoretical calculations and what is based on the actual TAC measured values? Has this methodology been applied to other cores and is there any ways that it can be verified by an alternative method? I have to confess that I got lost here… Clearer information should be provided.

We see that we have to explain better how the melt fraction is estimated. For present day 100% melt will not result in 0 cm3 g-1 air content. We calculate in the supplement how much this is theoretically assuming total equilibration between water and atmosphere. TAC for 0% melt can be calculated or measured from recent samples. A linear relationship between those two anchor points is used to calculate the melt fraction from TAC (presented in supplementals) with the assumption that TAC was constant throughout the Holocene. As temperature variations are low this is a reasonable assumption.

Temperature is then calculated using the HIRHAM model. (see our responses to this question further down)

Their reconstructions need to be better discussed in the context of the other existing reconstructions such as the ones shown in Figure 4 from Buizert et al. (2018). In the context of their temperature reconstruction and this applies to the whole interval, I think showing the water isotopic profile from RECAP is really missing. I am aware it is challenging to infer robust quantitative temperature reconstruction from water isotopes in Greenland but still it does provide some useful information (e.g. NEEM community members 2013) and it would be very valuable to use it in this context.

The RECAP water isotope record and temperature reconstruction based on it are not topic of this manuscript. We present our reconstruction together with other reconstructions in figure 4. This figure will be modified according to the request of the reviewer (see our answer further down).

I am also wondering about how large the melt layers are? How many TAC data are measured over a melt layer? Could an uncertainty bar be added to the graph in Figure S6?

Melt layers are not homogeneously distributed. Such an estimate would be completely arbitrary.

**Section 4.3 (relation to climate changes during DO events):**

I find it very difficult to follow the reasoning and what has been done here. The authors need to clarify greatly what they have done both in the main text and the supplement. Amongst other things, in the SM, there is a mention of setting a value of 0.9 to the midpoint of the CH4 increase, I don't understand this. Also, Figure S8 is way too small, it is not possible to see anything. How was the timing of the start of dust and CH4 increase defined? Was it done visually or what it done using a systematic or statistical method such as done in studies such as Rasmussen et al. 2014 or Erhardt et al. 2019?

In several places in the paper and in this section in particular, the authors mention the delta age changes across the ice core, however they do not provide any quantitative information about it nor about how it has been estimated. More quantitative details should be provided and it would also be good to have a figure showing its evolution. This information could be discussed in the section about the RECAP timescale in the SM.

Dage can be estimated at the start of DO events from the depth difference when $CH_4$ and $\delta^{18}O$ of $H_2O$ increase. We realize that Dage is variable in that core. That is why we ignore it as there is not enough data allowing for a systematic investigation.

**Section 4.3 (TAC and insolation):**

It is unclear to me whether the authors are using the ISI380 curve as the local insolation curve solely because this is the curve used for the EDC TAC record or whether they also proceeded to the evaluation of the threshold by tuning the precession to obliquity amplitude ratio of ISI in the power spectra on the corresponding ratio of TAC and they effectively found that at RECAP 380 W.m-2 was also the appropriate threshold. Note for instance that at NGRIP, Eicher et al. (2016) established that the most appropriate curve for ISI 390. I would like also to point out that in a recent paper, Raynaud et al. (2023) evidence a strong correlation between TAC and the mean half year summer insolation and propose that the latter is a more appropriate orbital curve to use than the ISI380. Indeed, as mentioned previously, the ISI threshold was evaluated by tuning the precession to obliquity amplitude ratio of ISI in the power spectra on the corresponding ratio of TAC. A limitation is that it involves some circular reasoning, as in turn, because of that resemblance ISI380 was used for orbital tuning purposes. Hence, I would suggest the authors to look into the relationship between RECAP TAC data and the mean half year summer insolation instead, the added value is that it is chosen independently from a tuning on the TAC power spectra and it also allows for the change of length of seasons.

As discussed in this section, we essentially find no correlation between local insolation and RECAP TAC. This will not change no matter which of the currently popular definitions ISI380, ISI390, mean summer half year is used.

**Section 4.4:**

Considering the large uncertainties attached to the estimate of past elevation of the RECAP site, I would suggest the authors to rephrase their results such as the elevation at the RECAP site did not change significantly between the LGM and today. And hence, their results are coherent with the prime hypothesis that the Renland ice cap did not change elevation through time. Indeed, the authors need to be rigorous in the phrasing of the results: what they did

apply to the RECAP site but it is important to stay cautious when extrapolating to the state of the Renland ice cap.

We are happy to rephrase as suggested.

**Conclusions:**

Please be more quantitative especially regarding the summary of the paleoelevation during the LGM reconstructions. Provide also some perspectives because I feel that a lot of topics are touched upon and it leaves many open questions and opens a lot of doors for subsequent studies. This should be highlighted.

We agree with the reviewer, many questions are open concerning TAC. We will rephrase the conclusions and also the abstract as also requested by reviewer 1.

Finally, the authors need to be careful when revising their manuscript: In places, the text is redundant between the main manuscript and the appendix or supplementary material. For instance, the text found between line 278 and line 285 is also found in the supplementary material. Also the text in Appendix A, line 366 to 370 is a copy of the text present in the main manuscript (including a reference to the Appendix itself!).

For the readers convenience, we deliberately repeat information in the supplementals that is also given in the main text. (we will of course remove the reference to itself)

**Other comments:**

Line 12: Avoid the use of the term Eemian, and also be more specific that it is only a portion of the Last interglacial that is covered considering that it is represented by the interval 119-121 ka.

Eemian is an established term for that time period and we prefer to continue using it.

Line 36: Revise the title of the section, it is more about the RECAP ice core drilling site than the Renland ice cap; Also in this section, it would be good the add a few sentences related to the position of this drilling site in relation to the old Renland ice core site.

We cannot see how this is relevant to the topic of this manuscript. However, we will add reference to the 1988 drill site.

Line 40: What do the authors mean by temperature measured in the firn? Is this estimate an average on the whole firn column? Or is it a measurement done at the surface? Please specify this and how it was measured (presence of a weather station for monitoring over a few years?).

The firn temperature below the reach of annual temperature oscillations is representing the annual mean temperature. This is also what we write.

Line 45: Revise the numbering of the supplementary figures, this is the first time the authors refer to a figure and it is numbered S3. Also, add a space between 530 and m.

We prefer to keep the numbering of supplementals as is. It follows the logic of first describing the system then the data. Renumbering would be confusing.

Line 49: On Figure 1: to add the location of the previous Renland ice core.

We will be adding as much information as we can about the relative position of the RECAP core with respect to the 1988 core.

Line 94: Ref to Raynaud et al. 2007 is not appropriate in this context.

Correct, will be removed.

Line 96: Eicher et al. (2016) use a threshold at 390 W.m-2 and not 320 W.m-2, it needs to be corrected.

Typo will be corrected, thank you.

Line 173: It is unclear to me if the authors display their record on an ice age scale or a gas age sale, this should be specified.

Will be specified.

Line 185: The caption of Figure 2 states that it presents an estimated mean annual renland temperature but it seems to me that it is misleading. As far as I understand the only temperature reconstruction inferred from a Renland ice core is for the section covering the Holocene (Vinther et al. 2009). The older part of the curve is the NGRIP temperature scaled up by 13°K if I understand well because this is the temperature difference between the two sites at the Holocene or present-day ? But surely, this is not reasonable to assume that this offset has been constant through time? The authors need to clarify this and discuss this limitation somewhere. Also, I would suggest to temperature in °C rather than K as usually done in most if not all paleoclimate publications presenting temperature reconstructions, so it is more intuitive.

The reviewer understands correctly what we describe in the caption. We clearly state that what is presented is an estimate for the Renland mean annual temperature. We do the most conservative approach here which is a constant offset. Change to °C will be done.

Line 192: I don't find this statement correct: the different potential factors contributing to TAC variations in Greenland and Antarctica might be similar but I believe it is still an open question regarding their relative influence, it may change from one ice core site to the other (even within Greenland and Antarctica).

This is actually what we mean. We will formulate more precise.

Line 216: Figure 4. Add numbering for the panel and adjust the caption accordingly as well as clarify it. It is currently challenging to understand this figure, what is represented and where do the different variables come from. For instance, amongst other things, I don't understand the representation of the percentage of melt. Temperatures should also be provided in °C and ideally as anomalies relative to today.

We agree the data can be presented more clearly and we will do so.

Line 226: Has the model HIRHAM5 been validated at other sites regarding its capability to reconstruct summer temperature from melt information? Please provide information. Please provide information also related to the attached uncertainty and represent it on the figure.

We are not aware of any studies (with the HIRHAM5 model or others) that use melt information to infer temperatures. However, Langen et al., 2017 that we are referring to evaluated melt amounts and melt extents against in-situ and satellite-based observations.

Line 240: Change the title e.g. "TAC changes at the end of the Last interglacial" and please avoid the use of Eemian for the Last interglacial as strictly speaking this is not appropriate when referring to climate changes inferred in ice core records (see discussion in Govin et al. 2015). Also, the record doesn't quite fully cover the Last Interglacial but effectively only the end.

Happy to write "end of previous interglacial". However, we still prefer to use Eemian as an identifier for the time period. This is customary in the community although it is strictly speaking not correct as principally only referring to the Netherlands.

Line 260: Please clarify the caption of Figure 5 (dust record is in orange and continuous CH4 in red?)

We will remove the connecting red line for CH4 and leave the dots.

Line 264: Section 4.2 rather than Section 4.3?

Thank you.

Line 279: sentence starting with "As methane…" and the next one: Please provide reference to this statement. This is very unclear, please rephrase. Also you mention that delta age is very variable, plese provide quantified estimate and show the delta age evolution in a figure.

We will add a reference. As explained before, we see a further elaboration on RECAP Dage based on the limited data available as not appropriate.

Line 268: Please refer to the figures in the supplement here and consider moving these figures in the main text, they are interesting!

Assuming this comment refers to Line 286; the reference to the supplemental figures is already there.

Line 285: Eicher et al. is redundant, remove one of the two references.

Yes, thank you

Line 294: …than in the NGRIP ice core…

Done

Line 296: Figure 6 is not intuitive and easy to understand, please provide more information to guide the reader in understanding it. Also in the text, the authors do provide timing estimates (e.g. sentence staring line 192), so would it be possible to redo the figure with an x-axis being the time rather than this normalized age with no unit ?

We can explain better. Applying an age is not possible as it is variable across D-O events.

Line 300: Another section 4.3? Please correct the numbering of the section.

Thank you for noticing

Line 347: sea level.

Thank you!

Line 361: please state the actual quantified uncertainty here.

This is detailed in section 4.4

Line 367: It is unclear to me why is the modeling work presented in an appendix and not in the SOM like the other supplementary information.

The model calculation is central for our interpretation of the Holocene and Eemian part of the record. Therefore we prefer to keep it in the main manuscript.

**Supplementary material:**

Page 6, S6: Following up on a comment I made above, the authors should consider mentioning the delta age evolution and showing a figure of it in this section.

This is very involved and would be patchy. Also see our response further up.

Page 7, S7.2: to add a caption to the table.

Will be added.

Page 15, S10: the authors mention GS18, GS19.1 and GI23.1. If they want to leave this information, they should highlight them on the Figure S10. However, it doesn't seem to me that it is relevant since they don't really discuss these intervals elsewhere.

Will be removed.

**References**:

Bazin et al. CP, 9, 1715, 2013.

Bouchet et al., CP, 19, 2257 ; 2023

Eicher et al. CP, 12, 1979, 2016

Erhardt et al. CP, 15, 811, 2019.

Epifanio et al. TC, https://doi.org/10.5194/tc-17-4837-2023

Govin et al. QSR, 129, 1, 2015.

NEEM project members, Nature, 493, 2013

Rasmussen et al. QSR, 106, 14, 2014.

---

## Author Response (AR1)

Total Air Content measurements from the RECAP ice core
Sindhu Vudayagiri, Bo Vinther, Johannes Freitag, Peter L. Langen, and Thomas Blunier
**Handling editor**: Amaelle Landais, amaelle.landais@lsce.ipsl.fr

**Our point-by-point response to the two reviewers' comments below.**

**RC1**

**Referee comment for: 'Total Air Content measurements from the RECAP ice core', Vudayagiri et al.**

This manuscript presents Total Air Content (TAC) measurements throughout the full RECAP ice core from the Renland ice cap, Greenland. The initial aim was to understand the elevation history of the ice cap, as TAC of ice core bubbles can be a proxy for ice elevation given the relationship between elevation and barometric pressure. However, much of the measured record was affected by either melt induced lowering of air content values, or rapid climatic transitions dominating the TAC signal due to non-elevation induced effects on pore volume. The main interpretations therefore presented are a melt correction for the TAC in the Holocene section of the core, and an elevation calculation for a short section of the core in the Last Glacial Maximum which is presumed unaffected by melt or strong climatic changes.

Much analysis has been conducted in this work, and it would be of benefit to see these results published. It is nice to see TAC measurements from two laboratories each with established and robust methodologies, which can demonstrate analytical comparability, especially given the more complex outcomes of the data. The authors have commendably looked for additional interpretation of the data given that much of it could not be used for elevation.

For me, the structure and writing of the paper lacked clarity, making it difficult to follow. The rational of the paper seemed somewhat unclear – the $d^{18}O$ record suggested as an anchor for the hypothesis of no elevation change at Renland isn't presented or discussed further, and previous studies of Greenland TAC (NGRIP) have shown climatic influences affecting TAC values so I am not clear on how an elevation history was expected to be obtained throughout the whole of RECAP. I am also missing what the wider scientific conclusions of the paper are, for example could there be some discussion on the application of TAC measurements to Greenland cores given the observations in this paper? Everything needs pulling together more succinctly for the reader. In some areas, I would like to see further detail, for example with the age scale and with errors on the different analytical systems used. I give more specific comments below.

We fully agree with the reviewer that most of the manuscript discusses the obstacles to interpreting TAC in terms of altitude. Only during the LGM do we believe that the RECAP data allow such a calculation. We have now clarified this in the abstract and conclusions. We see this manuscript as a step towards further research to understand TAC, which we believe will remain an important parameter in the climate puzzle in the near future.

There is a misunderstanding concerning the $\delta^{18}O$ of $H_2O$ data. Data from Renland (older Renland core from 1988) has been used in Vinther et al., 2009. It is not presented here. We explain this in the revised version.

**Major questions:**

Abstract:

The abstract somewhat peters out, not effectively summarising the paper, and I suggest it is rewritten. Given the main goal of the paper was for an elevation study, why does it not give the elevation estimate for the one section of the core which allowed it? The final two sentences state two things that were done in the paper without presenting the result of them. There are no final remarks on the scientific takeaways of this paper. Improving this may help the clarity of the rest of the paper.

Abstract has been rewritten.

Introduction:

Line 28 states that 'the study uses d$^{18}$O of H$_2$O from the Renland ice cap as an anchor point arguing that the ice cap has not experience significant elevation changes'. However, I could not see this record referenced here or presented in the paper. This should be included. A recent paper by Grieman et al., 2024 present an example of the use of d$^{18}$O in conjunction with TAC for elevation change, published I note after submission of this paper but a useful reference for the revision. (https://doi.org/10.1038/s41561-024-01375-8).

The statement refers to the study of Vinther et al., 2009. We have essentially removed that section now only using it to explain the initial motivation for our study. The Grieman study is interesting but refers to a different situation and approach and we see no need to cite it here.

Section 1.2:

This section would benefit from subheadings to break the text in to defined sections. The discussion on insolation is long for a general overview of the idea; can this be refined? I also think this discussion needs to be connected back to the expected or observed influence of insolation on the RECAP core itself to keep it relevant.

Also taking into account remarks from reviewer 2 this section has been massively shortened and condensed.

Line 105: It is referenced that the TAC record in the NGRIP core was influenced by D-O events and the conclusion was that climatically stable periods are preferential for elevation estimates. Therefore, why was the rational of the TAC measurements from RECAP to show elevation history of Renland throughout the core, when it would also likely show these same climatic influences over significant periods? Was it expected that you would observe something different?

When we started the study, we were not aware of the complications and the goal has been to confirm that the elevation was stable from the last glacial to today. Especially the short term millennial fluctuations came as a complete surprise to us. We have reformulated in the introduction to make this clear.

1. Measurements:

TAC from PSU was measured as part of d$^{15}$N analyses. Given that d$^{15}$N can be a useful indicator for firn column changes, could this available d$^{15}$N be helpful in the interpretation? It looks as though lower TAC values coincide with higher d$^{15}$N in the Supplementary Information plots. The plotted d$^{15}$N is not mentioned in the main paper, only a data repository link is given – can the main paper refer to these plots, or even include the d$^{15}$N in a plot in the main paper? I think there is interest here.

Over D-O events, the main influence on $\delta^{15}$N is thermal fractionation outcompeting changes in the firn column. Separating the two effects requires another parameter like argon isotopes. $\delta^{15}$N is useful to determine when transitions start which we use and already mention in the manuscript.

What are the errors of the respective analytical systems used at PSU and PICE? I can see final pooled standard deviations of the data but not the errors of the analytical systems themselves.

The analytical errors are below the natural variability of TAC. In essence, it is the precision of the pressure gauge and temperature measurements.

2. Results and Discussion:

The TAC data are reported to be presented on the RECAP GICC05 age scale, but the reference for this seems to be incorrect; the Simonsen 2018 paper does not appear to discuss an age scale. Please replace with appropriate citation or included some detail on the age scale in this paper.

The reviewer is correct, this is the wrong reference. Fixed.

Is the existing RECAP GICC05 age scale an ice age scale only, or did it include an existing gas age scale? I am not clear on whether this paper has constructed their own gas age scale. If the latter, can the gas age scale please be presented somewhere? For clarity, it would also be helpful if it could be stated somewhere which data are presented on which of these two age scales.

We use the time scales published in Simonsen et al., 2019. For most plots the difference between ice age and gas age is not visible. Information on the time scale used has been added to all figures. Also, all data are presented on the ice time scale.

3. Conclusions:

I would like to see the concluding section bring together the investigations of the paper more effectively. I am missing final remarks on the wider scientific impact of this paper – for example given the results of this paper, what are the suggestions for future TAC analysis in other Greenland cores?

We added concluding remarks as requested.

**Minor/technical points:**

Abstract: Suggest changing TAC 'is' a proxy to TAC 'can be' a proxy, in the complications of the proxy and the context of the results of this paper.

Reformulated: "In principle, TAC is…"

Line 23: Can you say why Renland ice cap elevation stability is a pertinent question?

Generally, this is not questioned but we believe it is better to confirm that assumption.

Line 39: at an elevation 'of' 2315 m a.s.l. … near 'the' summit.

Done

Line 54: total air content has been defined already so can just be TAC, also missing a comma after TAC.

Comma added

Line 71: Can you clarify whether the parameter you use is the one with or the one without the later additional sites?

It is already stated that we use the one without the additional sites on line 70.

Line 105: This it the first mention of Dansgaard-Oeschger events in the main text I think, define 'D-O' here since it is used herein.

Done

Line 189: 'The temperature effect will be below 1%' – can you explain or reference this value?

This results from the ideal gas law. We added a line explaining this.

Line 191: (in) TAC

Done

Line 199: Line scan(ning)

Line scan is the term that is understood and used in the community.

Line 203: Line scan(ner)

Line scan is the term that is understood and used in the community.

Figure 4: The top step-plot of melt % does not appear to have an axis. Please amend this.

The figure has been improved based on above and other comments of both reviewers.

Lines 225 – 228: This has been directly copied and pasted from the Appendix which it refers to (or vice versa, since I notice the same section in Appendix A still self-refers to Appendix A. Please reword one to avoid repetition and remove self-reference in Appendix.

Fixed

Numbering of subsections is incorrect after 4.2.

Fixed

Line 279-286: Again this section is directly copy and pasted from/to the Supplement so needs re-writing. In both sections, 'Why this is the case we ignore' seems quite a negative phrasing to me, perhaps it just a preference but I would ask to change this. You don't explain this further because you don't believe it impacts your results, or because you do not know the explanation?

We do not have an explanation. It is very strange compared to other cores. We reformulated that section.

Line 307: 'outlined' rather than 'lined out'

Done

Supplement:

The supplement needs a thorough proofread by the authors for the present grammar/typing errors.

Fixed

S1: Is the bubble size determined by eye looking at the photos with the scale? If so, does this have quite a large error?

The bubble diameters are measured with a calliper. This information has been added to the section. The uncertainty of the bubble diameter is substantial by natural variability and the measurement uncertainty. Figure S3 (S5 in the revised version) shows standard errors for the bubble diameters on bag means (55cm sections).

Figure S8: Difficult to read this figure at the size presented.

Size has been increased.

'DO-events' is used in the supplement, while it is 'D-O events' in the main text. Should be the same for consistency.

Fixed

**RC2**

The manuscript by Vudayagiri et al. present a new Total Air Content (TAC) record measured on the RECAP ice core from Greenland and covering the past ~121 000 years. The authors combine TAC dataset measured in two laboratories. The first dataset has been measured using a new experimental set-up dedicated to TAC analyses at PICE, Niels Bohr Institute (Denmark) and the second dataset has been obtained indirectly using the existing experimental set-ups for $CH_4$ and $\delta^{15}N$ of $N_2$ analyses at Penn State University (USA). The authors show that it is challenging to use the new TAC record to infer robust information about past elevation changes at RECAP. Still they make an attempt for the last Glacial Maximum and they follow other research directions to extract climatic information from their new record (e.g. melt fraction and amplitude of the warming during the mid-Holocene and at the end of the Last Interglacial) and investigate the controlling factors of both the short- and long- timescale variations in the RECAP TAC record (e.g. firn structure changes, local insolation).

I'm surprised (to say the least) about the shape of this manuscript especially considering the list of experts who worked on it. Indeed, the current manuscript requires some very major revisions both in terms of the form and the content in order to reach a state that is reasonable enough so it can be considered for publication. In fact, I think that it is a real pity that this manuscript is in such poor shape because it is nonetheless very nice to see a new TAC record measured at high resolution on a costal site in Greenland. I think it is important that this new record is eventually published. I also think that the different research directions based on this new records which are presented (but never really fully developed) are interesting and well within the scope of Climate of the Past.

We would like to thank the reviewer for their effort and their extensive and very detailed review. They are obviously knowledgeable about the topic, and we value their time and effort to work through our manuscript which, we know, is not an easy read due to the complexity of TAC.

We agree that recent findings, including the ones presented in this manuscript, show that TAC is quite difficult to understand. There are a number of pitfalls, and we make that more clear in the conclusions and abstract as requested (also by reviewer one).

**Major comments:**

Overall, I found the current manuscript very difficult to read and to understand. I would argue that it is due to the fact that the manuscript contains a number of big problems both related to the form and the content. Indeed:

- o   I find the form and structure of the manuscript of poor quality. It is written as such that the logic followed for the development of the different paragraphs and the different sections is often hard to grasp, it often lacks of transitions between the paragraphs. In addition, the captions of the figures are not precise enough and the figures themselves often not big enough making hard to see what the authors are describing. I also believe that there are too much important information and calculations, and too many figures that have been placed in the supplementary material (SM), making the reader having to go constantly back and forth between the main manuscript and the SM, this is not ideal.

We did our best to make the text easier to follow. The issues with TAC in the RECAP core are clearly different between the Holocene/ previous interglacial and the glacial sections. The structure of the manuscript reflects that. This naturally leads to clear breaks. We believe that the nitty gritty (but important) details are all put to the supplementals where they belong allowing the reader to get the main message of the manuscript from the main text without distraction. From following up remarks it seems that what the reviewer is missing in the figure caption is the information about the time scale used (ice age or gas age). The difference is not visible, but we now specify which age scale is used which now is always the ice time scale.

- In many places, the work performed is either not well described and vague (in several places, background information is missing), or in opposite, sometimes, the information becomes very technical and no background information is provided to help the reader to follow, especially in the case that the reader is not a specialist of this climate proxy. All this makes it very hard to identify properly what the key results coming out of this study are.

We complemented the specific information requested under minor comments.

- Various research topics related to TAC are being tackled in the manuscript going from investigating the melt fraction based on TAC to reconstruct summer temperature, to using TAC during the Last Glacial Maximum (LGM) to provide information on paleo-elevations, to investigating the link of TAC changes at RECAP with insolation changes, and also how they are affected by the abrupt Dansgaard-Oeschger (DO) events. All these topics are worth looking into but maybe it is too much for one paper? If the authors want indeed to tackle them all, they need to do an important effort to guide thoroughly the readers by providing a more comprehensive introduction providing referenced background context, presenting in both a succinct and precise way the different aspects, announcing the outline of the different following sections etc. Alternatively, the authors could consider refocus their study on only a selection of topics. It could be beneficial to improve the clarity and structure of the paper and it would also prevent having such a large SM.

We prefer to publish all TAC results of the RECAP ice core in one manuscript. We believe that this is the best way to communicate the complexity of TAC in this core to the reader. We introduce the different issues with that core/record in the abstract and introduction and believe that it is now easier to follow.

- In turn, I would like to mention that as far as I understand this is the first time that the experimental system developed at PICE is presented in a publication. As a result, I believe that stronger emphasise should be put on the presentation and the assessment of the new system e.g. no clear description and quantitative estimates of the attached uncertainties is provided. To me, the description of this new experimental system is an important result of the study in itself and it would deserve also to be highlighted, for instance in the abstract and the conclusions of the study.

We added a sketch of the system and pictures of the extraction chambers to the supplementals where we also moved the description of the setup. Now the entire introduction of the system including calibration is in one place.

In the following I provide more detailed comments information related to these main remarks and suggestions for changes. I would strongly recommend the authors to consider them when preparing a new version of their manuscript.

**-Abstract:**

It needs to be rewritten in order to focus precisely on what is discussed/presented in the paper. Most sentences are very vague. The authors tell us what they investigated but not the actual results found. Here they should highlight for instance what is new compared to other TAC records, what are the quantitative results related to summer temperature reconstructions. Also, if the authors want to mention the paleoelevations estimates for the LGM, they need to also be more specific and provide the actual calculated estimates. It seems to me that the fact that a new experimental set-up for measuring TAC is important and should be also included in the main messages of this abstract.

==Abstract has been rewritten.==

**-Section 1.2, paragraph starting line 74:**

This section needs to be revised to follow a linear development as it is very hard to read. It currently jumps from mentioning the short-term variability of Vc to the role of solar insolation on $O_2/N_2$ and then the link between local insolation and TAC. This makes it very unclear and it lakes transitions between the paragraphs. A suggestion would be to (1) introduce the fact that there is a link on long timescales between local insolation and TAC, (2) present the parallel between TAC and the $O_2/N_2$ ratio and the potential physical mechanisms to explain the link of these tracers to local insolation, (3) next, to mention that based TAC has been used for constraining through orbital dating the latest Antarctic reference chronologies (AICC2012 and AICC2023, Bazin et al. 2012; Bouchet et al. 2023), and (4) then present and discuss the fact that TAC is also influenced by short-term climate variations. The authors cite the Eicher et al. (2016) paper but there is also the recent paper by Epifanio et al. (2023).

==The section has been shortened and restructured. Following reviewer 1, a subheading for perennial variations of TAC has been added. Epifanio et al, (2023) is referenced.==

**Section 2:**

As mentioned previously and as far as I understand this is the first time that the experimental system developed at PICE is presented in a publication. As a result, I believe that stronger emphasise should be put on the presentation of the system, e.g. it is necessary to add a schematic of the experimental system. Also, it is essential to provide a more substantial description and quantifications of the different sources of uncertainties attached to the measurements and to state the overall uncertainty attached to each analysis of TAC with the new system, e.g. the authors mention briefly the uncertainty related to the calibration of the experimental volume but there are other sources such as the one related to the water vapour pressure evaluation, the mass loss during the time of sample pre-evacuation or the uncertainty related to the measurement of the pressure gauge itself ?

==We moved the detailed description of the system to supplements so the main text concentrates on the data. In supplements, we provide now a sketch of the system which is==

==described in detail, and pictures of the extraction chambers. Requested uncertainties are given in the text. This is presented together with the calibration of the system, so the reader has one spot to get all the information about the new system.==

I understand that the PSU experimental protocol has been presented in details elsewhere, however the authors should still state the uncertainty attached to the TAC measurements performed with this system.

==As explained, the PSU system has not been built for TAC measurements. Therefore, we carefully compare their result to the PICE data in the supplementals. No systematic offset is found between the datasets.==

In the section 4, the authors provide a pooled standard deviation but (1) I don't think this is the place to provide this information (this should be done in section 2) and (2) I don't understand what it represents effectively and what it does and does not account for. This needs to be clarified.

==We introduced a section in between the introduction of the measurement system and the discussion of the data comparing the datasets. We also made it clear that this discussion is on the final datasets including all corrections.==

Finally, in my opinion, the presentation of this new system at PICE is an important added value to the paper and hence this could be better highlighted for instance in the abstract and the conclusion of the paper.

==Happy that the reviewer sees our system as an "important added value". However, we do not believe it that special that it needed more highlights. However, we added a sketch of the system to the supplementals. See also our comments higher up concerning the PICE TAC system.==

**Section 3:**

The dataset from PSU are corrected for the CBE using a linear regression defined based on the PICE data, this regression needs to be presented more clearly and discussed. I would suspect the depth-interval considered is going to have an impact on the slope of the regression line and hence the values of the CBE corrections for PSU measurements. Figure S3 presents some error interval but no details are provided regarding how this was quantified, this is missing and should be added.

==The uncertainty of the regression is already given in the supplemental text. We added the name of the matlab function that has been used to calculate it.==

The authors discuss in the supplementary material the agreement between the PICE and PSU dataset. I think this is a shame that it is placed there as this section is very important for the readers to gauge how far it is possible to go into the interpretation of the new records when combined. Also the current discussion is very brief and qualitative, it needs to be revised and more quantitative.

We believe this technical aspect of the datasets belongs into the supplementals. It is referenced in the main text with the main results, namely that the datasets are compatible.

**Section 4:**

The structure of this section needs to be revised. In particular, before going into the interpretation of the different sections of the record, it is essential that clear presentation and description of the dataset are provided e.g. in term of the amplitude of the signals, their variability and how they compare. And then, a structure of the rest of the section needs to be announced so the reader knows what to expect in the following of the section. The titles of the sub-sections 4.1, 4.2 and 4.3 should also be more specific than just the name of a time period. Also, please check the numbering of the sections, there are some mistakes. For instance, a second section 4.1 (Glacial) comes after section 4.2 (previous Interglacial). That section 4.1 Glacial seems too short to be a section of its own too?

We have added a section comparing the three datasets as section 4. Therefore, this section is now section 5. We have added an introduction paragraph describing the data. We have also renamed headers in section 5 and slightly restructured.

The section related to the identification of the melt layer needs to be re-written so it is easier to follow the reasoning of the authors. They also need to provide some background regarding the identification of the meltlayers and how it is possible to infer some temperature reconstructions from the melt fraction. In fact, how is the melt fraction estimated, what is based on theoretical calculations and what is based on the actual TAC measured values? Has this methodology been applied to other cores and is there any ways that it can be verified by an alternative method? I have to confess that I got lost here… Clearer information should be provided.

We have restructured the section in order to make clear how we calculate the melt fraction from TAC. We have added some information from the supplements and refer clearer to the supplemental information where necessary.

Their reconstructions need to be better discussed in the context of the other existing reconstructions such as the ones shown in Figure 4 from Buizert et al. (2018). In the context of their temperature reconstruction and this applies to the whole interval, I think showing the water isotopic profile from RECAP is really missing. I am aware it is challenging to infer robust quantitative temperature reconstruction from water isotopes in Greenland but still it does provide some useful information (e.g. NEEM community members 2013) and it would be very valuable to use it in this context.

We adapted figure 4 according to comments elsewhere. We now show all temperatures as deviations, compare to Buizert et al. (2018) and to the reconstruction based on water isotopes from Vinther et al. 2009. We discuss accordingly in the text.

I am also wondering about how large the melt layers are? How many TAC data are measured over a melt layer? Could an uncertainty bar be added to the graph in Figure S6?

Melt layers are not homogeneously distributed. Such an estimate would be completely arbitrary. We see no way of including this request.

**Section 4.3 (relation to climate changes during DO events):**

I find it very difficult to follow the reasoning and what has been done here. The authors need to clarify greatly what they have done both in the main text and the supplement. Amongst other things, in the SM, there is a mention of setting a value of 0.9 to the midpoint of the CH4 increase, I don't understand this. Also, Figure S8 is way too small, it is not possible to see anything. How was the timing of the start of dust and CH4 increase defined? Was it done visually or what it done using a systematic or statistical method such as done in studies such as Rasmussen et al. 2014 or Erhardt et al. 2019?

In several places in the paper and in this section in particular, the authors mention the delta age changes across the ice core, however they do not provide any quantitative information about it nor about how it has been estimated. More quantitative details should be provided and it would also be good to have a figure showing its evolution. This information could be discussed in the section about the RECAP timescale in the SM.

$\Delta$age can be estimated at the start of D-O events from the depth difference when $CH_4$ and $\delta^{18}O$ of $H_2O$ increase. We added a sentence clarifying this and a reference.

Concerning the remarks to the supplement. We have added $\Delta$age to Fig. 9a-d. We now state how we pick $\Delta$age for RECAP and explain better why using the original EGRIP $\Delta$age the methane mid transition point gets a value of 0.9 assigned and not 1.

Also, we have added uncertainty of $\Delta$age to Fig. 6 in the main text.

**Section 4.3 (TAC and insolation):**

It is unclear to me whether the authors are using the ISI380 curve as the local insolation curve solely because this is the curve used for the EDC TAC record or whether they also proceeded to the evaluation of the threshold by tuning the precession to obliquity amplitude ratio of ISI in the power spectra on the corresponding ratio of TAC and they effectively found that at RECAP 380 W.m-2 was also the appropriate threshold. Note for instance that at NGRIP, Eicher et al. (2016) established that the most appropriate curve for ISI 390. I would like also to point out that in a recent paper, Raynaud et al. (2023) evidence a strong correlation between TAC and the mean half year summer insolation and propose that the latter is a more appropriate orbital curve to use than the ISI380. Indeed, as mentioned previously, the ISI threshold was evaluated by tuning the precession to obliquity amplitude ratio of ISI in the power spectra on the corresponding ratio of TAC. A limitation is that it involves some circular reasoning, as in turn, because of that resemblance ISI380 was used for orbital tuning purposes. Hence, I would suggest the authors to look into the relationship between RECAP TAC data and the mean half year summer insolation instead, the added value is that it is chosen independently from a tuning on the TAC power spectra and it also allows for the change of length of seasons.

As discussed in this section, we essentially find no correlation between local insolation and RECAP TAC. This will not change no matter which of the currently popular definitions ISI380, ISI390, mean summer half year is used. Therefore, we see no need to further analyse the data to support our conclusion that the RECAP data seems basically unaffected by insolation effects.

**Section 4.4:**

Considering the large uncertainties attached to the estimate of past elevation of the RECAP site, I would suggest the authors to rephrase their results such as the elevation at the RECAP site did not change significantly between the LGM and today. And hence, their results are coherent with the prime hypothesis that the Renland ice cap did not change elevation through time. Indeed, the authors need to be rigorous in the phrasing of the results: what they did apply to the RECAP site but it is important to stay cautious when extrapolating to the state of the Renland ice cap.

We are not sure how the reviewer would like us to formulate. We believe that we formulate cautiously and honestly what we find.

**Conclusions:**

Please be more quantitative especially regarding the summary of the paleoelevation during the LGM reconstructions. Provide also some perspectives because I feel that a lot of topics are touched upon and it leaves many open questions and opens a lot of doors for subsequent studies. This should be highlighted.

We agree with the reviewer, many questions are open concerning TAC. We added a few phrases in conclusions (also requested by reviewer 1).

Finally, the authors need to be careful when revising their manuscript: In places, the text is redundant between the main manuscript and the appendix or supplementary material. For instance, the text found between line 278 and line 285 is also found in the supplementary material. Also the text in Appendix A, line 366 to 370 is a copy of the text present in the main manuscript (including a reference to the Appendix itself!).

We have removed redundancy. However, some information is deliberately repeated for convenience.

**Other comments:**

Line 12: Avoid the use of the term Eemian, and also be more specific that it is only a portion of the Last interglacial that is covered considering that it is represented by the interval 119-121 ka.

We give the time interval and write previous interglacial while also using Eemian which is an established term for that time period.

Line 36: Revise the title of the section, it is more about the RECAP ice core drilling site than the Renland ice cap; Also in this section, it would be good the add a few sentences related to the position of this drilling site in relation to the old Renland ice core site.

Title has been changed to "The RECAP ice core" as requested. Distance to the 1988 drill site is now mentioned.

Line 40: What do the authors mean by temperature measured in the firn? Is this estimate an average on the whole firn column? Or is it a measurement done at the surface? Please specify

this and how it was measured (presence of a weather station for monitoring over a few years?).

Measured at 20m depth. Information has been added.

Line 45: Revise the numbering of the supplementary figures, this is the first time the authors refer to a figure and it is numbered S3. Also, add a space between 530 and m.

We prefer to keep the numbering of supplementals as is. It follows the logic of first describing the system then the data. Renumbering would be confusing.

Line 49: On Figure 1: to add the location of the previous Renland ice core.

The 1988 drill location is 1.5 km away from the RECAP site. We added this information to the text.

Line 94: Ref to Raynaud et al. 2007 is not appropriate in this context.

Corrected, (the section has also been rewritten).

Line 96: Eicher et al. (2016) use a threshold at 390 W.m-2 and not 320 W.m-2, it needs to be corrected.

Corrected, thank you.

Line 173: It is unclear to me if the authors display their record on an ice age scale or a gas age sale, this should be specified.

We now specify the time scale on all graphs.

Line 185: The caption of Figure 2 states that it presents an estimated mean annual renland temperature but it seems to me that it is misleading. As far as I understand the only temperature reconstruction inferred from a Renland ice core is for the section covering the Holocene (Vinther et al. 2009). The older part of the curve is the NGRIP temperature scaled up by 13°K if I understand well because this is the temperature difference between the two sites at the Holocene or present-day ? But surely, this is not reasonable to assume that this offset has been constant through time? The authors need to clarify this and discuss this limitation somewhere. Also, I would suggest to temperature in °C rather than K as usually done in most if not all paleoclimate publications presenting temperature reconstructions, so it is more intuitive.

Based on other comments, we chose to replace the temperature curve with the lately published $\delta^{18}O$ data, the usual primary parameter from ice cores.

Line 192: I don't find this statement correct: the different potential factors contributing to TAC variations in Greenland and Antarctica might be similar but I believe it is still an open question regarding their relative influence, it may change from one ice core site to the other (even within Greenland and Antarctica).

This is actually what we mean. We changed this paragraph significantly and think it is now formulated clearly.

Line 216: Figure 4. Add numbering for the panel and adjust the caption accordingly as well as clarify it. It is currently challenging to understand this figure, what is represented and where do the different variables come from. For instance, amongst other things, I don't understand the representation of the percentage of melt. Temperatures should also be provided in °C and ideally as anomalies relative to today.

This graph has been changed according to the comment above and comments elsewhere.

Line 226: Has the model HIRHAM5 been validated at other sites regarding its capability to reconstruct summer temperature from melt information? Please provide information. Please provide information also related to the attached uncertainty and represent it on the figure.

We are not aware of any studies (with the HIRHAM5 model or others) that use melt information to infer temperatures. However, Langen et al., 2017 that we are referring to evaluated melt amounts and melt extents against in-situ and satellite-based observations.

Line 240: Change the title e.g. "TAC changes at the end of the Last interglacial" and please avoid the use of Eemian for the Last interglacial as strictly speaking this is not appropriate when referring to climate changes inferred in ice core records (see discussion in Govin et al. 2015). Also, the record doesn't quite fully cover the Last Interglacial but effectively only the end.

Happy to write "end of previous interglacial". However, we also use Eemian as an identifier for the time period. This is customary in the community although it is strictly speaking not correct as principally only referring to a specific site in the Netherlands.

Line 260: Please clarify the caption of Figure 5 (dust record is in orange and continuous CH4 in red?)

We replaced the dust record with the $\delta^{18}O$ record. We also remove the connecting red line for $CH_4$ and only left the dots. The figure caption has been changed accordingly.

Line 264: Section 4.2 rather than Section 4.3?

Thank you.

Line 279: sentence starting with "As methane…" and the next one: Please provide reference to this statement. This is very unclear, please rephrase. Also you mention that delta age is very variable, plese provide quantified estimate and show the delta age evolution in a figure.

Reference added, $\Delta$age is now shown in supplemental plots.

Line 268: Please refer to the figures in the supplement here and consider moving these figures in the main text, they are interesting!

Assuming this comment refers to Line 286; the reference to the supplemental figures is already there.

Line 285: Eicher et al. is redundant, remove one of the two references.

==Done==

Line 294: …than in the NGRIP ice core…

==Done==

Line 296: Figure 6 is not intuitive and easy to understand, please provide more information to guide the reader in understanding it. Also in the text, the authors do provide timing estimates (e.g. sentence staring line 192), so would it be possible to redo the figure with an x-axis being the time rather than this normalized age with no unit ?

==We added "time scales" for the duration of an event reaching close off for the two cores in Fig. 6.==

Line 300: Another section 4.3? Please correct the numbering of the section.

==Corrected==

Line 347: sea level.

==Corrected==

Line 361: please state the actual quantified uncertainty here.

==Uncertainty added to conclusions as requested.==

Line 367: It is unclear to me why is the modeling work presented in an appendix and not in the SOM like the other supplementary information.

==The model calculation is central for our interpretation of the Holocene and Eemian part of the record. Therefore, we prefer to keep it in the main manuscript.==

**Supplementary material:**

Page 6, S6: Following up on a comment I made above, the authors should consider mentioning the delta age evolution and showing a figure of it in this section.

==Δage has been added to the figures in supplements==

Page 7, S7.2: to add a caption to the table.

==Added.==

Page 15, S10: the authors mention GS18, GS19.1 and GI23.1. If they want to leave this information, they should highlight them on the Figure S10. However, it doesn't seem to me that it is relevant since they don't really discuss these intervals elsewhere.

==Removed.==

**References**:

Bazin et al. CP, 9, 1715, 2013.

Bouchet et al., CP, 19, 2257 ; 2023

Eicher et al. CP, 12, 1979, 2016

Erhardt et al. CP, 15, 811, 2019.

Epifanio et al. TC, https://doi.org/10.5194/tc-17-4837-2023

Govin et al. QSR, 129, 1, 2015.

NEEM project members, Nature, 493, 2013

Rasmussen et al. QSR, 106, 14, 2014.

---

## Author Response (AR2)

Total Air Content measurements from the RECAP ice core
Sindhu Vudayagiri, Bo Vinther, Johannes Freitag, Peter L. Langen, and Thomas Blunier
**Handling editor**: Amaelle Landais, amaelle.landais@lsce.ipsl.fr

We thank the two anonymous reviewers for their second review. Our point-by-point response to the reviewers' further comments below.

Submitted on 04 Oct 2024

Anonymous referee #1

**Suggestions for revision or reasons for rejection**

(visible to the public if the article is accepted and published)

This manuscript has changed substantially following major revision, and the authors have clearly put significant time and effort in to improving the manuscript following the reviewer comments.

The manuscript is much clearer to follow, with a logical rational and structure. This makes the scientific takeaways easier to glean and provides a more coherent story despite there being a few different approaches to the scientific interpretation within one paper. It would be nice to see this dataset published and I just have some more minor comments:

The last 2K years – the Holocene record of TAC is largely compromised by melt, but in the last 2K years there are virtually no melt layers seen. TAC variability in this section is still rather high, outside of what can be related to elevation change. Perhaps the authors could add a comment on the cause of this variability?

> Some of the samples in the top region of the core have been selected since they show melt. Therefore, the variability in figure 2 does not represent randomly picked samples in that region. We added a sentence explaining this in the figure caption.

Figure 1(b): Can the red location point be made slightly larger and labelled? It is quite difficult to see.

> Changed accordingly.

Line 96-97: Given that it is stated at this early stage in the manuscript that TAC measurements are only useful for elevation estimates in climatically stable periods,

could the authors add comment on whether they expected this to be different for their measurements at Renland?

Sentence has been added as requested.

2 Measurements: How have the two systems been quantified for measurement uncertainty, and can these values be presented here to compare? I.e. Error of the analytical system itself, as opposed to pooled St.Dev of the TAC measurements. I feel I am missing a quantitative comparison of the two different systems. Also see related comment for supplement.

We are not able to provide that information. However, the pooled standard deviation is a measure integrating all uncertainties of the system including also the natural variability of TAC. Therefore, the similarity of the pooled standard deviations of the three measurement series shows that they are equivalent.

Figure 4: I was confused by the negative melt percentages at first. Reading the supplementary methodology, I'm guessing this comes from the wide range of TAC values spanning 0% melt in the calibration such that many values in this range sit above the final calibration? Perhaps this could be explained somewhere just for clarity, supplement would be fine.

We added an explanatory sentence to the figure caption.

Line 236: 'Comparable to today 120 kry ago': suggest rephrasing, 'comparable between 120 kry ago and today'.

Done

Line 260: 'And comparison to' should be 'and compare to'.

Done

Line 264-265: 'The time period considered, corresponding to the time it takes for surface snow to arrive at close off is delta age'.

The sentence seems identical to the one in the manuscript. We unfortunately do not understand what the reviewer would like us to change.

Line 272-273: repeats closely to lines 263-264, rephrase to avoid repetition.

We rephrased lines 272-273.

Line 280: 'when D-O manifest as drop': 'when the D-O event manifests as a drop'?

Done

Line 297: 'the correlation of (the?) spline of ISI...'

Done

Line 303: 'allowing (exclusion of) data affected...'

Done

Section 5.5: I would like to see a figure of the last glacial maximum data included in this section since it is directly working with it for the elevation reconstruction.

We moved figure S12 from the supplements into the main text. Accordingly, we deleted section S9 from the supplements. For readability we increased the font size in figure 7 (formally S12).

Line 337: 'For example' would be preferable to just 'E.g' for starting a sentence.

Done

Supplement

PICE TAC system – How accurate is the cutting of the samples, stated to be of 22 x 25 x 25 mm? Are they calliper measured and to what accuracy? Perhaps this fits in with the previous comment on measurement uncertainty being quantified for the two systems. Are potential errors such as this one included in quantification of measurement uncertainty?

Sample size is measured to 0.5mm precision. A sentence has been added to the supplementals. Such statistical errors are not included in quantification of the individual measurement uncertainty.

S2 final sentence: 'Given the relative to the sample....' This sentence is not very clear. Please rephrase.

Reformulated

S3 first sentence: ...'calculates from the'... remove 'from'.

The cut bubble effect (CBE) is indeed calculated **from** the average bubble diameter. Removing "from" would change the meaning of that sentence.

Figure S4: It is really quite difficult to see the bubbles in the ice that are being referred to, is there a clearer image available?

The provided picture is representative of the images that were used in the analysis.

Submitted on 22 Oct 2024

Anonymous referee #2

**Suggestions for revision or reasons for rejection**

(visible to the public if the article is accepted and published)
I have read the revised version of the manuscript by Vudayagiri and colleagues. I thank them for considering my comments and overall making their paper easier to follow and to understand. However, I still think that further few improvements are needed before it can be published. I list here additional comments I would ask the authors to consider.

I'm still a little puzzled by the beginning of the paper with the first introductive section (lines 25-36); it is more of a second abstract than a paragraph to introduce the subject of the manuscript. It is fine to keep this first paragraph short but the authors should consider providing more simply some introductive sentences about the use of TAC historically as an elevation change proxy (referencing studies from Raynaud and colleagues) and keeping their text of the motivation to use it for the RECAP core considering the previous studies by Vinther et al. and the previous Renland ice core. This is fine to then list the different topics tackled in the paper but I would suggest it to present it as an outline of the rest of the manuscript e.g. in the following Section X will tackle Y. In section 3, we will investigate etc.

> We thank reviewer 2 for the feedback. We appreciate the suggestions regarding the introductory section. However, we prefer to keep the current structure and content of the introduction as it is. We believe it effectively sets the stage for the manuscript by summarizing the motivation and scope without delving too deeply into historical details.

In general, I still find that there are still left some imprecise statements and title sections in several places. I would urge the authors to go through their manuscript to fix that. I'm not listing them all here. But for instance, they should be careful when referring at ice core names, it should be specified when it is about the ice core, or when it is about the TAC data, examples are at lines 26 (it should be Renland ice core), line 183 (it should be the Renland ice core site) or line 264 (it should be the NGRIP and RECAP TAC records).

> We changed accordingly for the first and last example. The statement on line 183 is true for entire Renland so not only the coring site. Therefore, we would like to leave it as is.

-Line 10: to reformulate such as: "The RECAP TAC data shows incoherently low values during time intervals corresponding to the Holocene climatic optimum (6 to 9 kyr b2k) and part of the last interglacial (119 to 121 kyr b2k) originating from melt layers which renders the TAC data unfit for paleo elevation interpolation.

> Done

-Line 20: to reformulation such as: "Within uncertainty, the elevation of the RECAP site during the last glacial maximum was similar to today.

Done

-Line 26: to reformulate to : "…used the first Renland ice core (+ add references)".

Done

-Line 33: to remove the term Eemian in brackets and elsewhere in the manuscript. I read the answers from the authors however 1/ I don't agree that it is customary in the ice core community to use that term (it was just initiated through one prominent publication presenting the NEEM ice core) and hence 2/ I still see no added value to use the term "Eemian" to refer to the Last interglacial when investigating the Last interglacial period in ice core records. I would again refer the authors to the paper Govin et al. QSR 2015, a community-led effort to try and avoid, amongst other things, confusions to be brought with the misuse of certain terms defined in specific archives to designate time intervals such as the Last interglacial.

I, Thomas Blunier, do not agree. Throughout my career, the term "Eemian" has been consistently used to refer to the previous interglacial in Greenland. While the added value may be debatable, we will remove the term from the manuscript for good measure.

-Line 36: Define the LGM acronym here and use it in the rest of the manuscript.

Done

-Line 78: to reformulate the sub-section title to "TAC variations at orbital –scale"

Done

-Line 93: to reformulate the sub-section title to "TAC variations at millennial-scale". I agree that rapid TAC variations are not fully understood however the papers cited by the authors (Eicher et al. 2016 and Epifanio et al. 2023) do propose some possible hypotheses related to changes in the firn structure. This should be formulated here with a couple of additional sentences.

We believe the title is adequate since the changes seem to occur also on shorter than "millennial scale" and prefer to leave it as is.

We have shortened this introductory section on the reviewers' previous requests. We believe that as an introduction stating "lacking understanding" is a fair statement.

-Line 164: I would re-iterate my comment to avoid a section title that is just a time interval (here "holocene" but also later with "the last interglacial"), the authors should propose more specific titles.

Changed to "The RECAP TAC Holocene record".

-Line 209: In their answer to the review, the authors mention that in Langen et al. (2017), modelled melt amounts and melt extents have been evaluated against in-situ and satellite-based observations. I believe that this would be a valuable information to add here.
Also, line 210: rephrase "extrapolated temperatures" into "modelled temperature"

Added in appendix A.

-Line 249: I think I now understand better what is the purpose of this section and it is helpful to have the figures shown bigger to look at the results. However, I think that the authors could still improve its presentation by formulating properly the objectives of the approach developed. "To create a general picture of what is happening in the firn column" imprecise (and also redundant statement as it is in line 263 and then line 272.
Also, they formulate the following results:" For both cores, on average, the TAC values start to decrease around the depth (time) when CH4 starts to increase at the beginning of a D-O event. However, the minimum TAC is found before the depth (time) when D-O manifest as drop in dust or increase in d18O. For NGRIP this minimum is reached some 600 years before the snow associated with the D-O event reaches close off while for RECAP it is about 150 years." But they don't propose an interpretation nor discuss potential ways to investigate this further (for instance using specific experiments with firn densification models). I find that it is missing.

The second statement has been adapted on the request of reviewer 1.
Firn air modelling, if it is to be helpful, requires a hypothesis that we currently are not able to offer. If reviewer 2 has suggestions on the processes that might be at work, we are happy to hear them.

-Line 292: Reformulate the title to "RECAP TAC and local summer insolation". Also, regarding the link between orbital-scale TAC changes and local summer insolation: I understand that changing the ISI target to ISI390 or some other type of local summer insolation curve will not change the result that orbital-scale TAC changes are not significantly correlated but the authors should still at least acknowledge in the text that there are some open questions related to the most appropriate choice of orbital curve to use for comparison with TAC variations. The authors could refer to the recent work from Raynaud et al. CP 2024 which is discussing this in details.

The title has been updated as requested. However, we do not see a compelling reason to include a reference to the latest paper by Raynaud et al., as we are simply describing our own approach in this context.

-Line 304: Reformulate the title to "Elevation change reconstructions from RECAP TAC during the Last Glacial Maximum"

Done

-Line 345: to reformulate the sentence as such: "…and toward the end of the last interglacial (119 kyr to 121 kyr b2k)

Done